# Entropic Time Schedulers for Generative Diffusion Models

**Dejan Stančević** *
Radboud University

**Florian Handke**
Ghent University

**Luca Ambrogioni**
Radboud University

## Abstract

The practical performance of generative diffusion models depends on the appropriate choice of the noise scheduling function, which can also be equivalently expressed as a time reparameterization. In this paper, we present a time scheduler that selects sampling points based on entropy rather than uniform time spacing, ensuring that each point contributes an equal amount of information to the final generation. We prove that this time reparameterization does not depend on the initial choice of time. Furthermore, we provide a tractable exact formula to estimate this *entropic time* for a trained model using the training loss without substantial overhead. Alongside the entropic time, inspired by the optimality results, we introduce a rescaled entropic time. In our experiments with mixtures of Gaussian distributions and ImageNet, we show that using the (rescaled) entropic times greatly improves the inference performance of trained models. In particular, we found that the image quality in pretrained EDM2 models, as evaluated by FID and FD-DINO scores, can be substantially increased by the rescaled entropic time reparameterization without increasing the number of function evaluations, with greater improvements in the few NFEs regime. Code is available at `https://github.com/DejanStancevic/Entropic-Time-Schedulers-for-Generative-Diffusion-Models`.

## 1 Introduction

Generative diffusion models (Sohl-Dickstein et al., 2015), and especially score-based diffusion models, have achieved state-of-the-art performance in image (Dhariwal & Nichol, 2021; Rombach et al., 2022; Song et al., 2021) and video generation (Ho et al., 2022; Singer et al., 2022). Generative diffusion models are obtained by reverting a forward diffusion process, which injects noise into the distribution of the data until all information has been lost. In practice, the performance of these models is highly dependent on the choice of a noise scheduling function that regulates the rate of noise-injection (Song et al., 2022). In most commonly used models, a change of noise scheduling is mathematically equivalent to a change of time parameterization. From a theoretical perspective, the choice of time parametrization, or equivalently of noise scheduling, is not constrained by theory since any change of time in the forward process is automatically corrected in reverse dynamics (Song et al., 2021). However, as explained above, the choice of time is very important practically since it affects both the temporal weighting during training and the discretization scheme during inference. Consequently, an 'incorrect' choice of time variable can lead to severe inefficiencies due to the under-sampling of some temporal windows and the redundant over-sampling of others. This is particularly problematic since recent theoretical and experimental work suggested that 'generative decisions' tend to be clustered in critical time windows (Li & Chen, 2024), which have been connected to

---

*dejan.stancevic@donders.ru.nl

39th Conference on Neural Information Processing Systems (NeurIPS 2025).

symmetry-breaking phase transitions in physics (Raya & Ambrogioni, 2023; Ambrogioni, 2025; Biroli et al., 2024; Sclocchi et al., 2025). The "triviality" of the first phase of diffusion prior to the initial phase transitions has led to the idea that this early phase can be skipped in one 'jump' using a pre-trained initialization Lyu et al. (2022). These late initialization schemes can be seen as a special case of time re-scheduling that compresses the high-noise part of the original schedule.

The idea of changing the diffusion time in a data-dependent way, also known as time-warping, was first introduced in Dieleman et al. (2022) in the context of a class of diffusion models for sequences of discrete tokens. However, their implementation required the used of special architectures trained with cross-entropy loss instead of the standard denoising score matching. In this paper, we show that a natural data-dependent time parametrization can be tractably obtained for any continuous generative diffusion model as the (rescaled) conditional entropy of $\mathbf{x}_0$ given $\mathbf{x}_t$. This choice of time leads to a constant entropy rate, meaning that each time point contributes to the final generation in an equal amount. Furthermore, we show that this *entropic time* is invariant, meaning that it does not depend on the original choice of time parameterization. Examples of the same SDE in the entropic time and standard time are given in figure 1. Furthermore, inspired by the optimality results, we introduce a *rescaled entropic time*. We provide an exact tractable formula that relates these quantities to the empirical EDM (Karras et al., 2022) and DDPM (Song et al., 2022) loss, which can be used to easily define the entropic time for any given trained network.

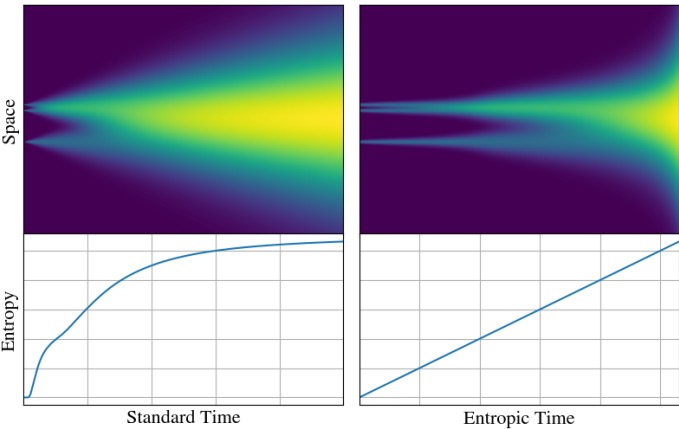

Figure 1: An example of the same SDE and its conditional entropy in the standard and entropic time.

## 2 Related work

**Accelerated Sampling Procedures** One of the most significant challenges in current diffusion models is the slow generative process. Since the introduction of the connection between the diffusion models and SDEs (Song et al., 2021), a wide array of research has aimed to address this issue by designing better numerical integrators. Some of the research in that direction includes the works of Liu et al. (2022) and Lu et al. (2022). An alternative line of research focuses on optimizing the sampling time schedule itself. Sabour et al. (2024) presents a principled approach to optimizing sampling schedules in diffusion models by aligning them with stochastic solvers, enabling higher efficiency. Wang et al. (2024) splits the generation process into three categories (acceleration, deceleration, and convergence steps), identifies imbalances in time step allocation, and introduces methods to address them, leading to faster training and sampling. Lee et al. (2024) uses spectral analysis of images to design a sampling strategy that prioritizes critical time steps, improving quality while reducing the number of steps. Li et al. (2023) explores joint optimization of time steps and architectures for more efficient generation without additional training. While much of this research focuses on learning or empirically determining optimal sampling schedules, our work provides a more theoretical perspective based on ideas from information theory. The closest work to ours is Dieleman et al. (2022), in which they use a cross-entropy loss to deduce a time-warping function for diffusion language models. However, our work differs since we analyze the standard diffusion models, where cross-entropy is not available. Furthermore, their expression for the entropy is not exact, as it implies an assumption of conditional independence of the tokens given the noisy state. On the other hand,

here we provide exact formulas that can be applied to any generative diffusion model trained with denoising score matching, both in the continuous and in the discrete regime.

**Connection Between Entropy, Information Theory, and Diffusion Models** Diffusion models are inherently tied to concepts from information theory, particularly in the context of denoising Gaussian noise, which is a fundamental operation in information-theoretic frameworks. This connection has inspired a growing body of work exploring the interplay between diffusion models and information theory. Premkumar (2024) investigates entropy-based objectives for learning more robust generative models. Kong et al. (2023a), Kong et al. (2023b), and Franzese et al. (2025) aim to provide a clearer understanding of diffusion models through an information-theoretic lens. Although these works explore the connection between information theory and diffusion models, and employ similar equations to ours, our focus diverges slightly. We use information theory as a guide to design better sampling algorithms. Work exploring a similar direction to ours is Li et al. (2025). However, they explore the conditional entropy between two consecutive time steps given a fixed discretization grid, while we look at the conditional entropy between the current time step and time zero in a way that is invariant under the change of time and discretization.

## 3 Background on score-matching generative diffusion

The mathematics of generative diffusion models can be elegantly formalized in term of stochastic differential equations (SDE). Consider a target distribution $p(\mathbf{x}_0)$ defined by a data source such as a distribution of, for example, natural images, sound waves, or linguistic strings. We interpret this data source as the initial distribution of a diffusion process governed by the SDE:

$$dX_t = \boldsymbol{f}(X_t, t)dt + g(t)dW , \tag{1}$$

where $dW$ is a standard Wiener process, $\boldsymbol{f}(X_t, t)$ is a vector-valued drift function, and $g(t)$ is a scalar volatility function, which regulates the standard deviation of the input noise. The marginal densities of the process can be obtained from the Fokker-Planck equation:

$$\partial_t p_t(\mathbf{x}_t) = \sum_{j=1}^{d} \partial_{x_j} \left( -f_j(\mathbf{x}_t, t) + \frac{g^2(t)}{2} \partial_{x_j} \right) p_t(\mathbf{x}_t) , \tag{2}$$

where $\partial_t$ is the partial derivative with the respect to time and $\partial_{x_j}$ is the partial derivative with respect to the $j$-th component of $\mathbf{x}_t$. We denote the forward "solution kernel" of the diffusion process as $p(\mathbf{x}_t \mid \mathbf{y})$, which is the solution of the Fokker-Plank equation for $p_0(\mathbf{y}) = \delta(\mathbf{y} - \mathbf{x}_0)$. The core idea of generative diffusion is to sample from $\mathbf{x}_0$ by initializing an asymptotic noise state $\mathbf{x}_T$ (where $T$ is large enough for the SDE to reach its stationary distribution) and by "inverting" the temporal dynamic. This can be done using the reverse SDE:

$$dX_t = \left( f(X_t, t) - g(t)^2 \tilde{\nabla} \log p(X_t) \right) dt + g(t) d\tilde{W} , \tag{3}$$

which can be proven to give the same marginal densities of eq. 1 when initialized with the appropriate stationary distribution, which is usually Gaussian white noise. We denote the reverse solution kernel of the reverse dynamics as $q(\mathbf{x}_0 \mid \mathbf{x}_t)$, which can be interpreted as the optimal denoising distribution. The data-dependent key component of the reverse dynamics is the so-called *score function*, which can be written as an expectation over the optimal denoising distribution:

$$\nabla \log p_t(\mathbf{x}_t) = \mathbb{E}_{q(\mathbf{x}_0 \mid \mathbf{x}_t)} \left[ \nabla \log p_t(\mathbf{x}_t \mid \mathbf{x}_0) \right] . \tag{4}$$

In most practical forms of generative diffusion, the score function is approximated using a deep network $\boldsymbol{s}_{\boldsymbol{\theta}}(\mathbf{x}_t, t)$, where the parameters $\boldsymbol{\theta}$ are optimized by minimizing an upper bound on the quadratic score-matching loss:

$$\begin{aligned} \mathcal{L}_{\mathrm{SM}}(\boldsymbol{\theta}) &\equiv \mathbb{E}_{p_0(x_0), t \sim \lambda(t)} \left[ \|\boldsymbol{s}_{\boldsymbol{\theta}}(\mathbf{x}_t, t) - \nabla \log p_t(\mathbf{x}_t)\|^2 \right] \\ &\leq \mathbb{E}_{p_0(x_0), p(x_t \mid x_0), t \sim \lambda(t)} \left[ \|\boldsymbol{s}_{\boldsymbol{\theta}}(\mathbf{x}_t, t) - \nabla \log p_t(\mathbf{x}_t \mid \mathbf{x}_0)\|^2 \right] \equiv \mathcal{L}_{\mathrm{DSM}}(\boldsymbol{\theta}) \end{aligned} \tag{5}$$

where $\lambda(t)$ is a density defined on the time axis. Note that $\mathcal{L}_{\mathrm{SM}}(\boldsymbol{\theta})$ and $\mathcal{L}_{\mathrm{DSM}}(\boldsymbol{\theta})$ differ only by a constant and therefore have the same gradients and optima. However, $\mathcal{L}(\boldsymbol{\theta})$ is substantially more tractable as it does not require samples from the unknown optimal denoiser $q(\mathbf{x}_0 \mid \mathbf{x}_t)$.

## 4 Optimal sampling schedule as a change of time

In this section, we revisit a result from Sabour et al. 2024 and notice some interesting features. Inspired by it, we formalize what we mean by the change of time.

Obtaining the analytical expression for the optimal sampling schedule is difficult and, in most practical cases, impossible. However, Sabour et al. 2024 shows that for the EDM noise schedule (Karras et al., 2022), the optimal sampling schedule for the ODE flow when data comes from a normal distribution with variance $c^2$ has an analytical expression. More precisely, the sampling schedule, $[t_{min}, t_1, ... t_{max}]$, that minimize the KL divergence is given by

$$\arctan\left(\frac{t_i}{c}\right) = \alpha_{min} + \frac{i}{N}(\alpha_{max} - \alpha_{min}) \tag{6}$$

where $\alpha_{min/max} = \arctan\left(\frac{t_{min/max}}{c}\right)$ (see theorem 3.1 in Sabour et al. 2024). It turns out that this schedule is also optimal for the deterministic DDIM (Song et al., 2022), which we show in Appendix C. Moreover, since the DDIM solver is invariant under time change (Lu et al., 2022), its optimal schedule remains invariant under time change, making it particularly suitable for comparing different time parameterizations.

This implies that even in the simple case, the optimal schedule depends on the data distribution. In addition, this result frames the optimization of a sampling schedule as a problem of time change. Rather than selecting timesteps differently for different numbers of sampling steps (e.g. EDM scheduler), theorem 3.1 shows that one should think of the sampling schedule as a transformation of time such that the sampling schedule becomes linear in the new time. Furthermore, in section 5.3, we will connect equation 6 with the conditional entropy production.

### 4.1 Change of time

The change of time in SDEs is a powerful technique used to simplify their analysis and solutions. By altering the time variable, the dynamics of the SDE can be transformed into a more manageable form. More information can be found in section $8.3$ in Lawler (2010).

**Definition 4.1.** *We say a function $\phi$ is a proper time change if it is continuous and strictly increasing.*

It can be shown that given a proper time change, $f$, and a random process, $X_t$, that solves the SDE $dX_t = f_t(X_t, t)dt + g_t(t)dW_t$ , then $Y_t = X_{\phi(t)}$ solves $dY_t = \dot{\phi}(t)f_t(Y_t, \phi(t))dt + \sqrt{\dot{\phi}(t)}g_t(\phi(t))dW_t$ . Guided by the theory of time change, we define an equivalence between SDEs.

**Definition 4.2.** *Given two SDEs $dX_t = f(X_t, t)dt + g(t)dW_t$ and $dX_s = \tilde{f}(X_s, s)ds + \tilde{g}(s)dW$ , we say that they are equivalent up to a time change if there exists a proper time change, $\phi : t \mapsto s$, such that*

*1. $\dot{\phi}(t)\tilde{f}(x, \phi(t)) = f(x, t)$*

*2. $\sqrt{\dot{\phi}(t)}\tilde{g}(\phi(t)) = g(t)$.*

Furthermore, we can require $f(0) = 0$ without affecting anything (since it is equivalent to subtracting a constant from the original function). By requiring that, we get that a time change between two SDEs is unique if it exists. Under a time change, the forward kernels stay the same, in the sense that $p_t(x|x_0) = q_{\phi(t)}(x|x_0)$ holds (this follows from $Y_t = X_{\phi(t)}$). Essentially, time change squeezes and stretches the time axis but does not fundamentally change the diffusion process. Algorithm 1 shows how to implement sampling using time change.

Given this notion of equivalence, a natural question arises: Is there a preferred or canonical time parameterization? We argue that a conditional entropy, $\mathbf{H}[x_0|x_t]$, and quantities derived from it are good candidates. However, for $\mathbf{H}[x_0|x_t]$ to make sense, we assume that we are given an initial distribution, $p_0(x)$ (that is, a data set). Therefore, besides an SDE, we require a dataset for the entropic time. In the further text, we will always assume that the dataset is given and is the same for different time parameterizations of SDEs.

**Algorithm 1** Sampling using time change
___
1: **procedure** TIMECHANGESAMPLER($\{t_i\}_{i=0}^M$, $\{\phi(t_i)\}_{i=0}^M$, $\sigma(t)$, $s(t)$, $D_\theta(x, \sigma)$,
  solver($x, D_\theta, \sigma_{\text{cur}}, \sigma_{\text{next}}, s_{\text{cur}}, s_{\text{next}}$), $N$)
2:   $\tau_j \leftarrow \phi(t_0) + \frac{j}{N-1}(\phi(t_M) - \phi(t_0))$ for $j = 0, \dots N - 1$   ▷ Uniform spacing in new time
3:   $\tilde{t}_j \leftarrow \text{interp}(\tau_j; \{\phi(t_i)\}_i^M, \{t_i\}_i^M)$ for $j = 0, \dots N - 1$   ▷ Corresponding old time
4:   $\tilde{\sigma}_j, \tilde{s}_j \leftarrow \sigma(\tilde{t}_j), s(\tilde{t}_j)$ for $j = 0, \dots N - 1$
5:   $\tilde{\sigma}, \tilde{s} \leftarrow \{0, \tilde{\sigma}\}, \{1, \tilde{s}\}$
6:   **sample** $x \sim \mathcal{N}(0, \tilde{\sigma}_N^2 I)$
7:   **for** $j \in \{N, \dots, 1\}$ **do**
8:       $\sigma_{\text{cur}}, \sigma_{\text{next}} \leftarrow \tilde{\sigma}_j, \tilde{\sigma}_{j-1}$
9:       $s_{\text{cur}}, s_{\text{next}} \leftarrow \tilde{s}_j, \tilde{s}_{j-1}$
10:       $x \leftarrow \text{solver}(x, D_\theta, \sigma_{\text{cur}}, \sigma_{\text{next}}, s_{\text{cur}}, s_{\text{next}})$   ▷ e.g. Heun, DDIM, etc.
11:   **end for**
12:   **output** $x$
13: **end procedure**
___

# 5 Entropic time schedules

In this section, we introduce the concepts of entropic time and rescaled entropic time. First, we provide some reasons for using the conditional entropy as a new time parameterization. Then, we show how to obtain the conditional entropy in practice and show its connection with commonly used quantities in diffusion literature. Furthermore, we demonstrate that the entropic time parameterizations are well-defined and invariant under the initial time parameterization of the SDE. There are several possible choices for the entropy function, which highlight different aspects of information transfer. The most straightforward choice is the information transfer $T_t$. Consider an initial source $\mathbf{x}_0 \sim p_0$ is transmitted through a noisy channel $p(\mathbf{x}_t \mid \mathbf{x}_0)$, which is determined by the solution of the SDE given in eq. 1. The noise-corrupted signal is received and decoded using $q(\mathbf{x}_0 \mid \mathbf{x}_t)$. The amount of information transferred at time $t$ can be quantified as the difference between the prior and posterior entropy:

$$\mathbf{T}_t = \mathbf{H}[\mathbf{x}_0] - \mathbf{H}[\mathbf{x}_0|\mathbf{x}_t] = \mathbf{I}[\mathbf{x}_0; \mathbf{x}_t] \tag{7}$$

where $\mathbf{H}[\mathbf{x}_0] = \mathbb{E}_{p_0(x_0)}[\log p(\mathbf{x}_0)]$ is the entropy of the source, $\mathbf{H}[\mathbf{x}_0|\mathbf{x}_t] = \mathbb{E}_{p(x_0, x_t)}[\log p(\mathbf{x}_0|\mathbf{x}_t)]$ is the conditional entropy under the optimal denoising distribution, and $\mathbf{I}[\mathbf{x}_0; \mathbf{x}_t]$ is a mutual information. Therefore, it is natural to interpret this quantity as the amount of information available at time $t$ concerning the identity of the source data. Up to a constant shift, this is equivalent to using the time variable $\phi(t) = \mathbf{H}[\mathbf{x}_0|\mathbf{x}_t]$ in the forward process. This time axis is defined by having a constant conditional entropy rate between the final generated image and the noisy state at time $t$.

## 5.1 Characterizing the conditional entropy

Having established that a conditional entropy makes sense as a new time parameterization, a question arises: How do we calculate it in practice? In general, conditional entropy can be written as $\mathbf{H}[\mathbf{x}_0|\mathbf{x}_t] = \mathbf{H}[\mathbf{x}_0] - \mathbf{I}[\mathbf{x}_0; \mathbf{x}_t] = \mathbf{H}[\mathbf{x}_0] + \mathbf{H}[\mathbf{x}_t|\mathbf{x}_0] - \mathbf{H}[\mathbf{x}_t]$.

In practice, $\mathbf{H}[\mathbf{x}_t|\mathbf{x}_0]$ is easy to get once the forward kernel is known, but it is difficult to obtain a numerical value of $\mathbf{H}[\mathbf{x}_t]$. However, by looking at a time derivative of the conditional entropy, we get a method for obtaining a numerical value. The time derivative is given by

$$\dot{\mathbf{H}}[\mathbf{x}_0|\mathbf{x}_t] = \dot{\mathbf{H}}[\mathbf{x}_t|\mathbf{x}_0] - \dot{\mathbf{H}}[\mathbf{x}_t]. \tag{8}$$

Hence, to know the time derivative, we need to calculate the time derivative of $\mathbf{H}[\mathbf{x}_t]$. In case when an SDE is given by 4.1, the entropy production is given by

$$\dot{\mathbf{H}}[\mathbf{x}_t] = \mathbb{E}_{p_t(x_t)}[\nabla(f_t)] + \frac{g_t^2}{2}\mathbb{E}_{p_t(x_t)}[||\nabla \log p(\mathbf{x}_t)||^2]. \tag{9}$$

The equation is a well-known expression in nonequilibrium thermodynamics for entropy production (Premkumar, 2024). The derivation of the expression can be found in the appendix B. Similarly, we can obtain the similar expression for $\mathbf{H}[\mathbf{x}_t|\mathbf{x}_0]$. Combining these two expressions, we obtain

$$\dot{\mathbf{H}}[\mathbf{x}_0|\mathbf{x}_t] = \frac{g_t^2}{2}\left(\mathbb{E}_{p(x_t, x_0)}[||\nabla \log p(\mathbf{x}_t|\mathbf{x}_0)||^2] - \mathbb{E}_{p_t(x_t)}[||\nabla \log p(\mathbf{x}_t)||^2]\right). \tag{10}$$

Note that this expression depends on the data distribution only through the Euclidean norm of the score function, which is approximated by a neural network in diffusion models.

## 5.2 Estimating the entropy rate from the training loss

In this section, we present a connection between the conditional entropy rate and training loss. For more details on the derivation of these results, see the Appendix F. In practice, most diffusion models can be written using the framework introduced in Karras et al. (2022). In this framework, the SDE is written as $dX_t = \frac{\dot{s}(t)}{s(t)} X_t dt + s(t)\sqrt{2\dot{\sigma}(t)\sigma(t)} dW$, with $p(x_t|x_0) = \mathcal{N}(x_t; s(t)x_0, s(t)^2\sigma(t)^2 I)$ as a forward kernel. This leads to the following conditional entropy production

$$\dot{\mathbf{H}}[\mathbf{x}_0|\mathbf{x}_t] = \frac{D\dot{\sigma}(t)}{\sigma(t)} - s(t)^2\dot{\sigma}(t)\sigma(t)\mathbb{E}_{p_t(x_t)}[\|\nabla \log p_t(\mathbf{x}_t)\|^2] \tag{11}$$

where $D$ is a dimension of the space (e.g. for the MNIST dataset, it would be $28^2$). In the rest of this paper, we will be using this framework. The squared error, $\epsilon_t^2$, encapsulates our uncertainty at time $t$ about the final sample $x_0$ and is given by

$$\epsilon_t^2 = \mathbb{E}_{p_t(x_t)}[\mathbb{E}_{p(x_0|x_t)}[\|\mathbf{x}_0 - \mathbb{E}_{p(y_0|x_t)}[\mathbf{y}_0]\|^2]] = \mathbb{E}_{p_t(x_t)}[tr(\sigma_{\mathbf{x}_0|\mathbf{x}_t}^2)]. \tag{12}$$

Using the fact that we can write $\sigma_{\mathbf{x}_0|\mathbf{x}_t}^2$ as $\sigma(t)^2(I + s(t)^2\sigma(t)^2 H[\log p_t(x_t)])$ (see Appendix G), we get

$$\dot{\mathbf{H}}[\mathbf{x}_0|\mathbf{x}_t] = \frac{\dot{\sigma}(t)}{\sigma(t)^3}\epsilon_t^2. \tag{13}$$

Recognizing that $\dot{SNR} = -\frac{\dot{\sigma}}{\sigma^3}$ and $\dot{\mathbf{H}}[\mathbf{x}_0|\mathbf{x}_t] = -\dot{\mathbf{I}}[\mathbf{x}_t; \mathbf{x}_0]$, equation 13 is precisely the well-known $I$–MMSE identity from information theory (Guo et al., 2005).

Furthermore, integrating over time yields an expression for the conditional entropy:

$$\mathbf{H}[\mathbf{x}_0|\mathbf{x}_1] = -\int_0^1 \dot{SNR}(t)\, \epsilon_t^2\, dt. \tag{14}$$

This expression coincides with the continuous-time (infinite-step) limit of the variational lower bound derived by Kingma et al. (2021), revealing a direct information-theoretic characterization as already hinted by Kong et al. (2023a).

The previous results provide a simple way of estimating the conditional entropy rate from the standard loss function of a trained diffusion model due to a close connection between the squared error and the loss. This provides a tractable way to estimate the conditional entropy from the training error. Note that, using the error of the model entails an approximation since the entropy is defined with respect to the true score function and, therefore, does not take into account the discrepancy between the learned and true score.

To analyze this deviation, we start from a striking result: the conditional entropy production is, up to a multiplicative factor, the gap between the explicit and denoising score matching loss in 5! In fact, following the steps from Vincent (2011) and keeping track of the terms that are constant in $\boldsymbol{\theta}$, we have

$$\mathcal{L}_{\text{SM}}(\boldsymbol{\theta}) = \mathcal{L}_{\text{DSM}}(\boldsymbol{\theta}) - \mathbb{E}_{p(x_0,x_t),\lambda(t)}[\|\nabla \log p(\mathbf{x}_t|\mathbf{x}_0)\|^2 - \|\nabla \log p(\mathbf{x}_t)\|^2]. \tag{15}$$

Using expression 10, we can rewrite the above equality as

$$\mathcal{L}_{\text{DSM}}(\boldsymbol{\theta}) = \mathcal{L}_{\text{SM}}(\boldsymbol{\theta}) + \mathbb{E}_{\lambda(t)}\left[\frac{2}{g_t^2}\dot{\mathbf{H}}[\mathbf{x}_0|\mathbf{x}_t]\right]. \tag{16}$$

This relation can also be expressed at a single time point $t$ as

$$\dot{\mathbf{H}}[\mathbf{x}_0|\mathbf{x}_t] + \frac{g_t^2}{2}\delta_t^2(\boldsymbol{\theta}) = \frac{g_t^2}{2}\mathbb{E}_{\mathbf{x}_t,\mathbf{x}_0}\left[\|\boldsymbol{s}_{\boldsymbol{\theta}}(\mathbf{x}_t,t) - \nabla \log p_t(\mathbf{x}_t \mid \mathbf{x}_0)\|^2\right], \tag{17}$$

where $\delta_t^2(\boldsymbol{\theta}) = \mathbb{E}_{p_t(\mathbf{x}_t)}\left[\|\boldsymbol{s}_{\boldsymbol{\theta}}(\mathbf{x}_t,t) - \nabla \log p_t(\mathbf{x}_t)\|^2\right]$ denotes the mean squared error between the true score and its neural approximation. The right-hand side corresponds to our estimate of the conditional entropy production. It follows that the estimated entropy production always upper-bounds the true value, with the gap determined by the disagreement between the learned and true scores, $\delta_t^2(\boldsymbol{\theta})$. In this sense, $\dot{\mathbf{H}}[\mathbf{x}_0|\mathbf{x}_t]$ can be interpreted as the *irreducible* contribution to the loss, reflecting the intrinsic uncertainty of the optimal denoising process.

## 5.3 The entropic and rescaled entropic times

Here, we introduce a rescaled entropy and show that both rescaled entropy and conditional entropy are proper changes of time and are invariant under different time parameterizations of SDE. Proofs can be found in the Appendix D. First, we notice that in the case of continuous data, the conditional entropy goes to negative infinity at time equal to zero. In practice, this is not observed since diffusion models always start from a non-zero initial time. However, it adds arbitrariness to the overall curve of the conditional entropy. To combat this problem, guided by the observation that the change of time for the optimal sampling schedule for normally distributed data, eq. 6, is equal to the rescaled entropy (see Appendix E), we introduce a rescaled entropy as $\int_0^t \sigma(\tau)\dot{\mathbf{H}}[\mathbf{x}_0|\mathbf{x}_\tau]d\tau$. Algorithm 2 shows how to estimate rescaled entropy in practice (and how it was estimated in this work).

**Theorem 5.1.** *Given an SDE and initial data distribution $p_0(x)$, $\phi(t) = \mathbf{H}[x_0|x_t]$ and $\phi(t) = \int_0^t \sigma(\tau)\dot{\mathbf{H}}[\mathbf{x}_0|\mathbf{x}_\tau]d\tau$ are proper time changes.*

We call these time parameterizations an *entropic time* and *rescaled entropic time*, respectively. Naturally, an important question emerges: How does the time parameterization of an initial SDE influence its reparameterized form? We show that an SDE written in entropic time is unique and does not rely on its initial parameterization. More precisely, given two SDEs equivalent up to a time change, the SDEs expressed in their respective entropic times are equivalent up to a time change, with the time change being the identity function (i.e. drift and noise terms of SDEs in entropic times are related by conditions 1. and 2. from definition 4.2, and are the same since the time derivative of the time change is one).

**Theorem 5.2.** *Given two SDEs as given in definition 4.2, and following time changes*

    *1. $\phi : t \mapsto s = f(t)$*

    *2. $\Phi_t : t \mapsto \mathbf{H}_t[\mathbf{x}_0|\mathbf{x}_t]$*

    *3. $\Phi_s : s \mapsto \mathbf{H}_s[\mathbf{x}_0|\mathbf{x}_s]$,*

*it follows that*

$$F := \Phi_s \circ \phi \circ \Phi_t^{-1} : \mathbf{H}_t[\mathbf{x}_0|\mathbf{x}_t] \mapsto \mathbf{H}_s[\mathbf{x}_0|\mathbf{x}_s]$$

*is a proper time change implementing the equivalence and is equal to the identity map, $F = id$.*

A similar result holds for the rescaled entropic time as well. Therefore, once reparameterized in entropic time (or rescaled entropic time), no matter the starting SDE time parameterization, drift and noise are always the same.

---

**Algorithm 2** Estimation of rescaled entropy, $\int \sigma(\tau)\dot{\mathbf{H}}[\mathbf{x}_0|\mathbf{x}_\tau]d\tau$

---

1: **procedure** ESTIMATERESCALEDENTROPY($D_\theta(x,\sigma)$, $\sigma(t)$, $s(t)$, $\{t_i\}_{i=0}^N$, $M$)
2:     **sample** $x_0^j \sim p_0$ for $j = 1, \ldots, M$
3:     $\mathbf{x}_0 \leftarrow [x_0^1, \ldots, x_0^M]$
4:     $\mathbf{R}_{i:N} \leftarrow \mathbf{0}$
5:     **for** $i \in \{0, \ldots, N-1\}$ **do**
6:         **sample** $\nu_j \sim \mathcal{N}(\mathbf{0}, I)$ for $j = 1, \ldots, M$
7:         $\boldsymbol{\nu} \leftarrow [\nu_1, \ldots, \nu_M]$
8:         $\mathbf{x}_{t_i} \leftarrow s(t_i)\mathbf{x}_0 + s(t_i)\sigma(t_i)\boldsymbol{\nu}$
9:         $\hat{\mathbf{x}}_0 \leftarrow D_\theta(\mathbf{x}_{t_i}, \sigma(t_i))$
10:         $\epsilon^2 \leftarrow \frac{1}{M}\sum_{j=1}^M \left\|\hat{\mathbf{x}}_0^j - \mathbf{x}_0^j\right\|^2$
11:         $\mathbf{R}_{i+1} \leftarrow \mathbf{R}_i + \frac{\dot{\sigma}(t_i)}{\sigma^2(t_i)}(t_{i+1} - t_i)\epsilon^2$              ▷ Riemann sum
12:     **end for**
13:     **output** $\mathbf{R}_{i:N}$
14: **end procedure**

---

### 5.4 Spectral rescaled entropic time

We based our definition of rescaled entropic time on optimality results for isotropic Gaussian distributions. However, these results do not account for how different directions in an anisotropic Gaussian influence the optimal schedule. From equation 10, we observe that the total entropy production can be interpreted as a sum of contributions from all basis directions, where the basis can be any orthonormal set (since only norms of scores affect the production). This interpretation corresponds to one specific way of weighting different directions. For image data and diffusion in pixel space, we explore an alternative in this paper: setting the rescaled entropy in each Fourier basis direction to be equal to 1 at the final time (i.e., giving an equal importance for each frequency), and then weighting them by their respective amplitudes. Theorems from the previous section still hold for the spectral rescaled entropy since they hold for each frequency (basis). An example of the resulting rescaled entropy across different frequencies is shown in figure 2.

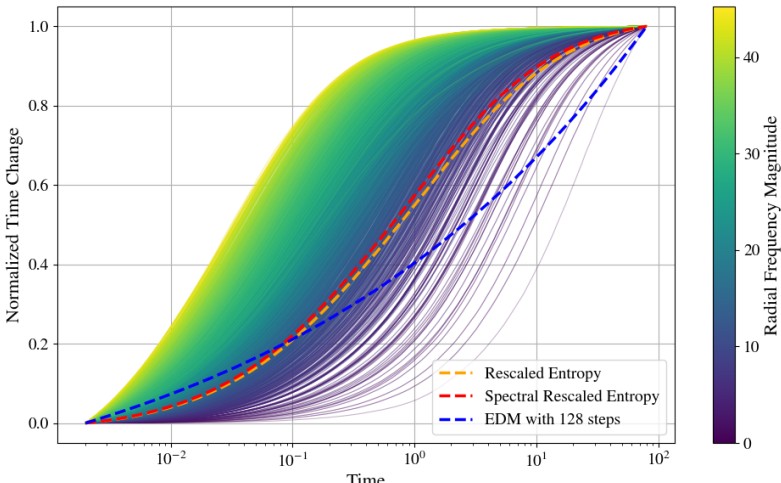

Figure 2: Normalized rescaled entropy as a function of radial frequency for the red channel in ImageNet-64, together with normalized rescaled entropy, spectral rescaled entropy, and EDM with 128 steps.

## 6 Experiments

We compare the performance of a few-step generation in the standard, entropic, and rescaled entropic times for several low-dimensional examples where an analytic expression for a score is easy to calculate. Next, we compare the performance of trained EDM and EDM2 models (Karras et al., 2022, 2024) on CIFAR10 (Krizhevsky et al., 2009), FFHQ(Karras et al., 2019), and ImageNet (Russakovsky et al., 2015) using the FID (Heusel et al., 2017) and FD-DINOv2 (Oquab et al., 2023; Stein et al., 2023) scores. More details about the setup can be found in the appendix H.

### 6.1 One-dimensional experiments

We used an analytic expression of a score function to compare the performance of a few-step generation process in different time parameterizations in one dimension. We used equidistant steps in the standard, entropic, and rescaled entropic times. We used the stochastic DDIM solver (Song et al., 2022). We compared those schedules for discrete data and a mixture of Gaussians. We used the Kullback-Leibler divergence to compare results for different schedules. An example of KL divergence behavior against the number of generative steps is given in figure 3. In general, we can see that in the discrete case, the entropic time outperforms other schedules by a large margin, while the standard schedule gives the worst results. Furthermore, we noticed that when variances of Gaussians are much smaller than the distance between them (i.e. there is no significant overlap between Gaussians), the entropic schedule gives better results. However, when the variances are not negligible in the mixture of Gaussians case, we can see that the rescaled entropic schedule gives the best results, while the

entropic schedule underperforms. This suggests that the entropic time might significantly improve certain discrete diffusion models.

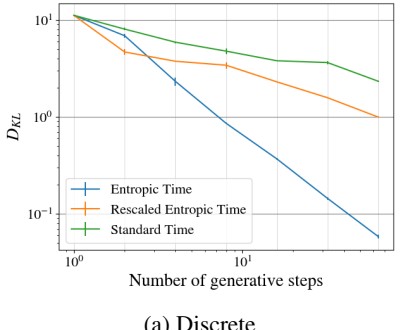
(a) Discrete

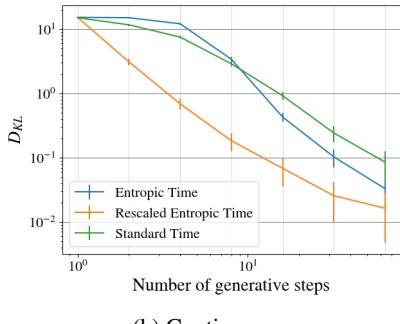
(b) Continuous

Figure 3: Kullback–Leibler divergence against the number of generative steps for different time parameterizations for mixture of 15 data points (discrete) and 15 Gaussians (continuous).

## 6.2 CIFAR10, FFHQ, and ImageNet

We compared the performance for different numbers of generative steps using standard, entropic, rescaled entropic, and spectral rescaled entropic (for diffusion in pixel space) times. To sample, we used the deterministic and stochastic DDIM solver. For CIFAR-10 and FFHQ, the EDM unconditional VP models were used (Karras et al., 2022). For ImageNet-64, the EDM2-S and EDM2-L models were used, while for ImageNet-512, the EDM2-XS and EDM2-XXL models were used. For ImageNet-512, we used both models: one optimized for FID and the other for FD-DINOv2.

Figure 4 gives an example of the effect of different schedules on generated images. We observed that the rescaled entropic schedule produces images with lower brightness. The results for FFHQ are presented in table 1, while the FID scores for FID-optimized networks and the DINOv2 scores for DINOv2-optimized networks on ImageNet-512 are reported in table 2. We note that the using the rescaled entropy schedule, model EDM-XS beats the FD-DINOv2 result pro-

Table 1: FID scores for different sampling schedules on FFHQ $64 \times 64$

| Solver | Schedule | FID ↓ | | |
|---|---|---|---|---|
| | | NFE=16 | NFE=32 | NFE=64 |
| Stochastic DDIM | EDM | 40.48 | 21.63 | 10.62 |
| | Rescaled Entropy | 30.81 | 14.89 | 7.60 |
| | Spectral Rescaled Entropy | **30.61** | **14.60** | **7.33** |
| Deterministic DDIM | EDM | 11.13 | 5.41 | 3.45 |
| | Rescaled Entropy | **8.10** | **4.28** | 3.16 |
| | Spectral Rescaled Entropy | **8.10** | **4.28** | **3.14** |

vided in Karras et al. (2024), 103.39, obtained using Heun solver. We observed that the entropic time produced unrecognizable images (see Appendix H), therefore, we have not included it in the results. The difference between spectral rescaled entropy and rescaled entropy is small but noticeable for stochastic DDIM, with spectral rescaled entropy performing better. In contrast, for deterministic DDIM, the difference is negligible. Results on CIFAR10 and FFHQ, together with more examples of generated images, are given in appendix H.

## 6.3 Limitations

While our results show a clear benefit of using the entropic schedules across a wide range of datasets and fast-sampling methods, we note that these benefits are observed specifically for the first-order solvers. We tried second-order solvers as introduced in Karras et al. (2022); Lu et al. (2022) but noticed worse results (compared to the EDM schedule). We believe this is due to the use of an inappropriate information transfer function, eq. 7. Specifically, the definition we use considers only the current time point, whereas second-order solvers also take into account the future time point when predicting the updated state, thereby altering the entropy rate. As a result, the mismatch in temporal perspective may lead to suboptimal performance for higher-order methods as their entropy curves probably need to be readjusted based on the features of the solver.

Table 2: FID and FD-DINOv2 scores for different sampling schedules for ImageNet-512

| Solver | Network | Schedule | FID ↓ | | | FD-DINOv2 ↓ | | |
|---|---|---|---|---|---|---|---|---|
| | | | NFE=16 | NFE=32 | NFE=64 | NFE=16 | NFE=32 | NFE=64 |
| Stochastic DDIM | EDM2-XS | EDM | 32.31 | 10.01 | 4.98 | 294.25 | 149.91 | 107.00 |
| | | Rescaled Entropy | **13.64** | **4.98** | **3.80** | **182.11** | **109.68** | **97.10** |
| | EDM2-XXL | EDM | 30.39 | 8.80 | 3.81 | 218.10 | 95.21 | 60.79 |
| | | Rescaled Entropy | **13.38** | **3.83** | **2.60** | **108.16** | **57.05** | **46.75** |
| Deterministic DDIM | EDM2-XS | EDM | 10.42 | 4.81 | 3.83 | **156.46** | **115.94** | 107.05 |
| | | Rescaled Entropy | **7.57** | **4.44** | **3.75** | 157.32 | 116.52 | **106.84** |
| | EDM2-XXL | EDM | 9.68 | 3.47 | 2.41 | 79.56 | 52.60 | 46.27 |
| | | Rescaled Entropy | **5.91** | **2.78** | **2.14** | **68.36** | **48.26** | **43.83** |

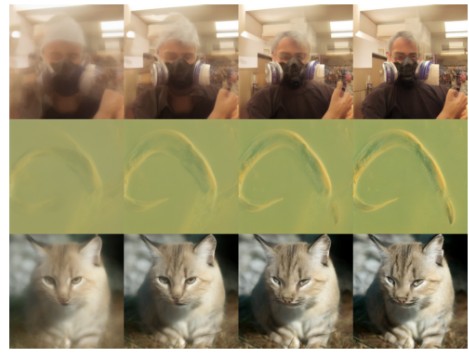 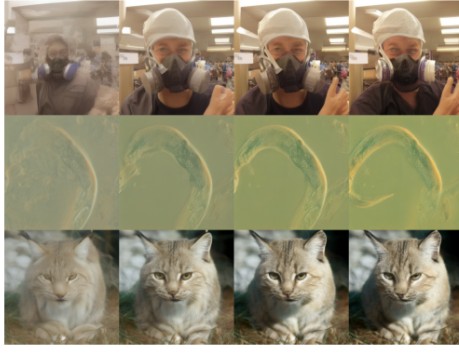

(a) EDM · (b) Rescaled Entropy

Figure 4: Comparison of generated images using EDM and rescaled entropic schedules with the same random seed. Images were generated using deterministic DDIM with NFE = 8, 16, 32, and 64.

# 7 Conclusions and future work

Several avenues for future work remain open. We conjecture that using conditional entropy provides an optimal schedule for discrete generative tasks, although we currently lack a theoretical proof; nonetheless, our toy examples showed great promise. Empirically aligning entropic time with discrete diffusion models, potentially in the spirit of time warping techniques such as Dieleman et al. (2022), is an exciting direction. Beyond this, entropic time may also offer a principled framework for training and model compression: in distillation, entropic time could identify the most informative stages for supervision and reduce redundancy in transferring knowledge from teacher to student models (similarly could be done for consistency models). More broadly, we propose entropic time as a candidate training schedule, enabling learning that is directly aligned with information flow. We are encouraged by the unexpected connection between our formulation and the continuous-time variational objective of Kingma et al. (2021). We envision that this perspective could eventually replace the heavy dataset-specific optimization required in approaches such as EDM (Karras et al., 2022), leading to more efficient and adaptive training across diverse modalities, including medical imaging, audio, and text. Lastly, we note that second-order solvers, which incorporate lookahead steps, may require a fundamentally different definition of information transfer. Developing entropic analogues tailored to such solvers is another important direction for extending this framework.

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

## A  Broader impact

This work proposes entropic and rescaled entropic time schedules for generative diffusion models, improving performance, especially in low NFE regimes, without additional training overhead. Our approach may benefit applications relying on fast generation, such as medical imaging and digital content creation. Furthermore, our work contributes to a deeper understanding of the relationship between information theory and generative modeling, which may inspire further theoretical advancements.

As with all generative models, there is potential for misuse in generating synthetic media that could be used for disinformation or impersonation. Improvements in efficiency may increase such risks by lowering the barrier to large-scale generation.

## B  Entropy Production

Here, we show

$$\dot{\mathbf{H}}[\mathbf{x}_t] = \mathbb{E}_{p_t(x_t)}[\nabla(f)] + \frac{g_t^2}{2}\mathbb{E}_{p_t(x_t)}[||\nabla \log p(\mathbf{x}_t)||^2]. \tag{18}$$

By looking inside the integral of $\dot{\mathbf{H}}[\mathbf{x}_t]$, we get

$$\begin{aligned}
\dot{\mathbf{H}}[\mathbf{x}_t] &= -\int \left( \dot{p}(x_t) \log p(x_t) + p(x_t)\frac{\dot{p}(x_t)}{p(x_t)} \right) dx_t \\
&= -\int \dot{p}(x_t) \log p(x_t) dx_t - \frac{d}{dt}\int p(x_t) dx_t \\
&= -\int \dot{p}(x_t) \log p(x_t) dx_t.
\end{aligned} \tag{19}$$

Assuming our dynamic is determined by the SDE 4.1, we can use the Fokker-Planck equation to simplify the derivative as follows

$$\begin{aligned}
\dot{\mathbf{H}}[\mathbf{x}_t] &= -\int \left( -\nabla \left( \left( f_t - \frac{g_t^2}{2}\nabla \log p(x_t) \right) p(x_t) \right) \right) \log p(x_t) dx_t \\
&= -\int \left\langle f_t - \frac{g_t^2}{2}\nabla \log p(x_t) \middle| \nabla \log p(x_t) \right\rangle p(x_t) dx_t \\
&= -\int \langle f_t | \nabla \log p(x_t) \rangle p(x_t) dx_t + \int \frac{g_t^2}{2} \langle \nabla \log p(x_t) | \nabla \log p(x_t) \rangle p(x_t) dx_t \\
&= -\int \langle f_t | \nabla p(x_t) \rangle dx_t + \int \frac{g_t^2}{2} ||\nabla \log p(x_t)||^2 p(x_t) dx_t \\
&= \int \nabla(f_t) p(x_t) dx_t + \frac{g_t^2}{2}\mathbb{E}_{p_t(x_t)}[||\nabla \log p(\mathbf{x}_t)||^2] \\
&= \mathbb{E}_{p_t(x_t)}[\nabla(f_t)] + \frac{g_t^2}{2}\mathbb{E}_{p_t(x_t)}[||\nabla \log p(\mathbf{x}_t)||^2]
\end{aligned} \tag{20}$$

which is exactly what we wanted to show. We used integration by parts in going from the first line to the second and from the fourth to the fifth.

## C  Optimal schedule for deterministic DDIM

Here, we show that the optimal schedule for the deterministic DDIM (Song et al., 2022) for the EDM SDE is the same as the one given in Sabour et al. (2024).

**Theorem C.1.** *Let $p_{data}(\mathbf{x}) = \mathcal{N}(0, c^2\mathbf{I})$ and the diffusion process is given by the EDM SDE. Sample $\mathbf{x}_{t_{\max}} \sim p(\mathbf{x}, t_{\max})$ and use the deterministic DDIM using $n$ steps along the schedule*

$$t_{\max} = t_n > t_{n-1} > \cdots > t_1 > t_0 = t_{\min}$$

*to obtain $\mathbf{x}_{t_{\min}}$. The optimal schedule $t^*$ minimizing the KL-divergence between $p(\mathbf{x}, t_{\min})$ and the distribution of $\mathbf{x}_{t_{\min}}$ is given by*

$$t_i^* = c \tan\left( \left(1 - \frac{i}{n}\right)\alpha_{\min} + \frac{i}{n}\alpha_{\max} \right)$$

*where*

$$\alpha_{\min} := \arctan(t_{\min}/c), \quad \alpha_{\max} := \arctan(t_{\max}/c).$$

*Proof.* The deterministic DDIM update is given as

$$\mathbf{x}_{t_{i-1}} = \hat{x}_0(\mathbf{x}_{t_i}) + \frac{t_{i_{i-1}}}{t_i}\left(\mathbf{x}_{t_i} - \hat{x}_0(\mathbf{x}_{t_i})\right)$$

where $\hat{x}_0(\mathbf{x}_{t_i}) = \mathbf{x}_t + t_i^2\nabla\log p(\mathbf{x}_{t_i})$. By using an analytical expression for the score, we get a simplified expression for the update

$$\mathbf{x}_{t_{i-1}} = \frac{c^2 + t_{i-1}t_i}{c^2 + t_i^2}\mathbf{x}_{t_i}.$$

This turns out to be exactly the same update as in ODE Euler method (equation 18 in Sabour et al. (2024)). Therefore, our claim follows from the proof of theorem 3.1. in Sabour et al. (2024). □

## D  Proofs from section 5.3

**Theorem D.1.** *Given an SDE and initial data distribution $p_0(x)$, $\phi(t) = \mathbf{H}[x_0|x_t]$ and $\phi(t) = \int_0^t ds\sigma_s\dot{\mathbf{H}}[\mathbf{x}_0|\mathbf{x}_s]$ are proper time changes.*

*Proof.* As already mentioned, a proper time change must be a strictly increasing, continuous function. Since $\mathbf{H}[x_0|x_t]$ has a derivative (see section 5.1), we need to show that it is positive. However, our claim follows from equation 13 (the squared error is equal to zero only when an initial distribution consists of one data point). □

**Theorem D.2.** *Given two SDEs as given in definition 4.2, and following time changes*

1. $\phi : t \mapsto s = f(t)$

2. $\Phi_t : t \mapsto \mathbf{H}_t[\mathbf{x}_0|\mathbf{x}_t]$

3. $\Phi_s : s \mapsto \mathbf{H}_s[\mathbf{x}_0|\mathbf{x}_s],$

*it follows that*

$$F := \Phi_s \circ \phi \circ \Phi_t^{-1} : \mathbf{H}_t[\mathbf{x}_0|\mathbf{x}_t] \mapsto \mathbf{H}_s[\mathbf{x}_0|\mathbf{x}_s]$$

*is a proper time change implementing the equivalence and is equal to the identity map, $F = id$.*

*Proof.* Immediately, we can see that $g$ is a proper time change since it is composed of other time changes. Similarly, using a chain rule, it is observed that $g$ implements the equivalence. Furthermore,

$$F(\mathbf{H}_t[\mathbf{x}_0|\mathbf{x}_t]) = (\Phi_s \circ \phi)(\Phi_t^{-1}(\mathbf{H}_t[\mathbf{x}_0|\mathbf{x}_t])) = \Phi_s(\phi(t)) = \mathbf{H}_s[\mathbf{x}_0|\mathbf{x}_{\phi(t)}]. \tag{21}$$

However, since $p_t(x) = q_{\phi(t)}(x)$ and $p(x_t|x_0) = q(x_{\phi(t)}|x_0)$, it follows

$$\begin{aligned}\mathbf{H}_t[\mathbf{x}_0|\mathbf{x}_t] &= -\iint p(x_t, x_0)\ln\left(p(x_0|x_t)\right)dx_0 dx_t \\ &= -\iint q(x_{\phi(t)}, x_0)\ln\left(q(x_0|x_{\phi(t)})\right)dx_0 dx_{\phi(t)} = \mathbf{H}_s[\mathbf{x}_0|\mathbf{x}_{\phi(t)}],\end{aligned} \tag{22}$$

where $x_t = x_{\phi(t)}$ (i.e. are the same spatial point) and time subscripts represent at which point in time the probability distribution is evaluated. This proves that $F = id$. □

Similarly, we can prove the same claim for the rescaled entropic time since $\sigma(t) = \sigma(\phi(t))$ for any proper change of time $\phi$.

# E    Rescaled entropy for Gaussian data

Here, we show that, in the case of the EDM noise schedule, the rescaled entropic time is the optimal sampling schedule for the ODE flow when data comes from a normal distribution with variance $c^2$ (equation 6).

Recall the expression for the rescaled entropy, $\int_0^t \sigma(\tau)\dot{\mathbf{H}}[\mathbf{x}_0|\mathbf{x}_\tau]d\tau$. From equation 11, we have

$$\int_0^t \sigma(\tau)\dot{\mathbf{H}}[\mathbf{x}_0|\mathbf{x}_\tau]d\tau = \int_0^t \left(D\dot{\sigma}(\tau) - s(\tau)^2\dot{\sigma}(\tau)\sigma(\tau)^2\mathbb{E}_{p_\tau(x_\tau)}[||\nabla\log p_\tau(\mathbf{x}_\tau)||^2]\right)d\tau. \tag{23}$$

Using the facts that $\sigma(\tau) = \tau$, $s(\tau) = 1$ and $\nabla\log p_\tau(x_\tau) = \frac{-x_\tau}{s(\tau)^2\sigma(\tau)^2 + s(\tau)^2 c^2}$, we get

$$\begin{aligned}\int_0^t \sigma(\tau)\dot{\mathbf{H}}[\mathbf{x}_0|\mathbf{x}_\tau]d\tau &= \int_0^t \left(D - \tau^2\frac{D}{\tau^2 + c^2}\right)d\tau \\ &= \int_0^t \frac{Dc^2}{\tau^2 + c^2}d\tau = Dc\arctan\left(\frac{t}{c}\right).\end{aligned} \tag{24}$$

Therefore, a linear sampling schedule, $[t_{min}, t_1, ... t_{max}]$, in the rescaled entropic time is given by

$$Dc\arctan\left(\frac{t_i}{c}\right) = Dc\left(\alpha_{min} + \frac{i}{N}(\alpha_{max} - \alpha_{min})\right) \tag{25}$$

where $\alpha_{min/max} = \arctan\left(\frac{t_{min/max}}{c}\right)$. Exactly the same as equation 6.

# F    Connection with a squared error and loss

In this Appendix, we show connections between conditional entropy production and some commonly used expressions in the diffusion literature. Firstly, we show how the conditional entropy production is related to the squared error at time $t$, $\epsilon_t^2$.

$$\begin{aligned}\epsilon_t^2 &= \mathbb{E}_{p(x_0,x_t)}[||\mathbf{x}_0 - \hat{\mathbf{x}}_0(\mathbf{x}_t)||^2] = \iint ||x_0 - \hat{x}_0(x_t)||^2 p(x_t|x_0)p(x_0)dx_t dx_0 \\ &= \iint \left|\left|x_0 - \frac{(x_t + s(t)^2\sigma(t)^2\nabla(\log p(x_t)))}{s(t)}\right|\right|^2 p(x_t|x_0)p(x_0)dx_t dx_0\end{aligned} \tag{26}$$

The squared error encapsulates our uncertainty at time $t$ about the final sample $x_0$. The following simplification of the above equation gives a more precise meaning.

$$\begin{aligned}\epsilon_t^2 &= \mathbb{E}_{p_t(x_t)}[\mathbb{E}_{p(x_0|x_t)}[||\mathbf{x}_0 - \mathbb{E}_{p(y_0|x_t)}[\mathbf{y}_0]||^2]] \\ &= \mathbb{E}_{p_t(x_t)}[tr(\sigma^2_{\mathbf{x}_0|\mathbf{x}_t})].\end{aligned} \tag{27}$$

From Appendix G, we know

$$Var_{p(x_0|x_t)}[\mathbf{x}_0] = \sigma(t)^2(I + s(t)^2\sigma(t)^2 H[\log p_t(x_t)]). \tag{28}$$

Hence,

$$\begin{aligned}\epsilon_t^2 &= \mathbb{E}_{p_t(x_t)}[tr(\sigma^2_{\mathbf{x}_0|\mathbf{x}_t})] = \sigma(t)^2\mathbb{E}_{p_t(x_t)}[tr(I + s(t)^2\sigma(t)^2 H[\log p(\mathbf{x}_t)])] \\ &= \sigma(t)^2(D - s(t)^2\sigma(t)^2\mathbb{E}_{p_t(x_t)}[||\nabla\log(p_t(\mathbf{x}_t))||^2]) \\ &= \frac{\sigma(t)^3}{\dot{\sigma}(t)}\left(\frac{D\dot{\sigma}(t)}{\sigma(t)} - s(t)^2\dot{\sigma}(t)\sigma(t)\mathbb{E}_{p_t(x_t)}[||\nabla\log(p_t(\mathbf{x}_t))||^2]\right) \\ &= \frac{\sigma(t)^3}{\dot{\sigma}(t)}\dot{\mathbf{H}}[\mathbf{x}_0|\mathbf{x}_t].\end{aligned} \tag{29}$$

Following notation from Karras et al. (2022) for the loss at time $t$, we have

$$\mathcal{L}(t) = \mathbb{E}_{p_0(x_0), \mathcal{N}(\epsilon; 0, I)}[\lambda(t) \|c_{out}(t)F_\theta - s(t)x_0 + c_{skip}(t)(s(t)x_0 + s(t)\sigma(t)\epsilon)\|^2]. \tag{30}$$

The formula for a prediction $\hat{x}_0(x_t)$ is given by

$$\hat{x}_0(x_t) = \frac{c_{out}(t)F_\theta(x_t) + c_{skip}(t)x_t}{s(t)}. \tag{31}$$

We can express the loss at time $t$ using the squared error as

$$\mathcal{L}(t) = \lambda(t)\mathbb{E}[\|s(t)\mathbf{x}_0 - s(t)\hat{\mathbf{x}}_0(\mathbf{x}_t)\|^2] = \lambda(t)s(t)^2\epsilon_t^2. \tag{32}$$

Furthermore, using the connection between a squared error and a conditional entropy production, we get

$$\dot{\mathbf{H}}[\mathbf{x}_0|\mathbf{x}_t] = \frac{\dot{\sigma}(t)}{\lambda(t)s(t)^2\sigma(t)^3}\mathcal{L}(t). \tag{33}$$

# G   Tweedie's second-order formula

Assume we are given a distribution $p(y)$ that is obtained by adding a Gaussian noise to a distribution $q(x)$, i.e. $q(y|x) = \mathcal{N}(y; sx, s^2\sigma^2)$.

Now given some $y \sim p(y)$, if we are interested in which $x \sim q(x)$ generated it, the best we can do is guess $\hat{x}(y) = E_{q(x|y)}[x]$. Tweedie's formula gives us

$$\mathbb{E}_{q(x|y)}[x] = \frac{y + s^2\sigma^2\nabla_y \log p(y)}{s} \tag{34}$$

Now, we might ask how sure we are of our guess. To answer that question, we need to look at the variance, $Var_{q(x|y)}[x] = \mathbb{E}_{q(x|y)}[x^2] - \mathbb{E}_{q(x|y)}[x]^2$. In this section, we derive the following result

$$Var_{q(x|y)}[x] = s\sigma^2\nabla_y E_{q(x|y)}[x] = \sigma^2(I + s^2\sigma^2 H[\log p(y)]). \tag{35}$$

However, a more general result regarding the cumulants of $q(x|y)$ holds (Dytso et al., 2022). That is, all the cumulants can be calculated using the score function and its derivatives.

Since we already have $\mathbb{E}_{q(x|y)}[x]$, we need to find an expression for $\mathbb{E}_{q(x|y)}[x^2]$.

$$
\begin{aligned}
\mathbb{E}_{q(x|y)}[x^2] &= \int dx \frac{q(y|x)q(x)}{p(y)}x^2 = \int dx \frac{q(x)x}{p(y)}q(y|x)x \\
&= \int dx \frac{xq(x)}{p(y)} \frac{yq(y|x) + s^2\sigma^2\nabla_y q(y|x)}{s} \\
&= \frac{y\mathbb{E}_{q(x|y)}[x]}{s} + \frac{s^2\sigma^2}{sp(y)}\nabla_y \int dx q(y|x)q(x)x
\end{aligned} \tag{36}
$$

Where in going from the first line to the second, we used $\nabla_y q(y|x) = \frac{sx-y}{s^2\sigma^2}q(y|x)$. However, we seem to have encountered a problem with the second term in our expression. However, by using $q(x, y) = q(y|x)q(x) = q(x|y)p(y)$, for the second term we get

$$
\begin{aligned}
\nabla_y \int dx q(y|x)q(x)x &= \nabla_y \int dx q(x|y)p(y)x \\
&= \nabla_y \left( p(y) \int dx q(x|y)x \right) = \nabla_y(p(y)\mathbb{E}_{q(x|y)}[x]) \\
&= \mathbb{E}_{q(x|y)}[x]\nabla_y p(y) + p(y)\nabla_y \mathbb{E}_{q(x|y)}[x].
\end{aligned} \tag{37}
$$

Hence,

$$
\begin{aligned}
\mathbb{E}_{q(x|y)}[x^2] &= \frac{y\mathbb{E}_{q(x|y)}[x]}{s} + \frac{s^2\sigma^2}{sp(y)}\left(\mathbb{E}_{q(x|y)}[x]\nabla_y p(y) + p(y)\nabla_y \mathbb{E}_{q(x|y)}[x]\right) \\
&= \frac{y\mathbb{E}_{q(x|y)}[x]}{s} + \frac{s^2\sigma^2\left(\mathbb{E}_{q(x|y)}[x]\nabla_y \log p(y) + \nabla_y \mathbb{E}_{q(x|y)}[x]\right)}{s} \\
&= \mathbb{E}_{q(x|y)}[x]\frac{y + s^2\sigma^2\nabla_y \log p(y)}{s} + \frac{s^2\sigma^2}{s}\nabla_y \mathbb{E}_{q(x|y)}[x] \\
&= E_{q(x|y)}[x]^2 + \frac{s^2\sigma^2}{s}\nabla_y \mathbb{E}_{q(x|y)}[x].
\end{aligned}
\tag{38}
$$

Now, we get an elegant expression for the variance

$$
Var_{q(x|y)}[x] = s\sigma^2 \nabla_y \mathbb{E}_{q(x|y)}[x] = \sigma^2(1 + s^2\sigma^2 \partial_{yy} \log p(y)).
\tag{39}
$$

So far, we have been dealing with one-dimensional random variables, but it is easy to generalize all the steps to arbitrary dimensions, which gives us the general formula

$$
Var_{q(x|y)}[x] = s\sigma^2 \nabla_y E_{q(x|y)}[x] = \sigma^2(I + s^2\sigma^2 H[\log p(y)]).
\tag{40}
$$

# H   Experimental details and Additional results

## H.1   Algorithms

Here, we present algorithm 3, which estimates the spectral decomposition of the squared error from the data. Having numerical values of the squared error and its spectral decomposition can be used to compute other entropic quantities of interest. A similar algorithm can be used to obtain entropy and squared error for different orthonormal basis.

---

**Algorithm 3** Estimation of spectral decomposition of $\epsilon^2(t)$

1: **Input:** $D_\theta(x,\sigma)$, Encoder$(x)$, $\sigma(t)$, $s(t)$, $t_{i\in\{0,\dots,N\}}$, $M$
2: **for** $i \in \{0,\dots,N\}$ **do**
3:      Sample $x_0^j \sim p_0$ for $j = 1,\dots,M$
4:      $\mathbf{z}_0 = $ Encoder$(\mathbf{x}_0)$
5:      Sample $\nu_j \sim \mathcal{N}(0, I)$ for $j = 1,\dots,M$
6:      $\mathbf{z}_{t_i} = s(t_i)\,\mathbf{z}_0 + s(t_i)\,\sigma(t_i)\,\nu$
7:      $\hat{\mathbf{z}}_0 = D_\theta\big(\mathbf{z}_{t_i}, \sigma(t_i)\big)$
8:      **for** $j = 1,\dots,M$ **do**
9:          $\Delta^j = \hat{z}_0^j - z_0^j$
10:          $\widehat{\Delta}^j = $ FFT2D$\big(\Delta^j\big)$
11:          $|\widehat{\Delta}^j|^2 = \big|\Re(\widehat{\Delta}^j)\big|^2 + \big|\Im(\widehat{\Delta}^j)\big|^2$       ▷ Modulus squared for each matrix entry
12:      **end for**
13:      $\mathcal{E}^2(t_i) = \frac{1}{M}\sum_{j=1}^M |\widehat{\Delta}^j|^2$       ▷ Matrix of squared error for different frequencies
14: **end for**
15: **output** $\mathcal{E}^2(t)$

---

## H.2   One-dimensional experiments

We used an analytic expression of a score function to compare the performance of a few-step generation process in different time parameterizations in one dimension. We used equidistant steps in the standard time, entropic time, and rescaled entropic time. All entropic quantities were obtained from the squared error using equation 13. The squared error was estimated at $10^4$ equidistant timesteps with $10^3$ samples at each timestep. We used a mixture of data points (discrete case) and a

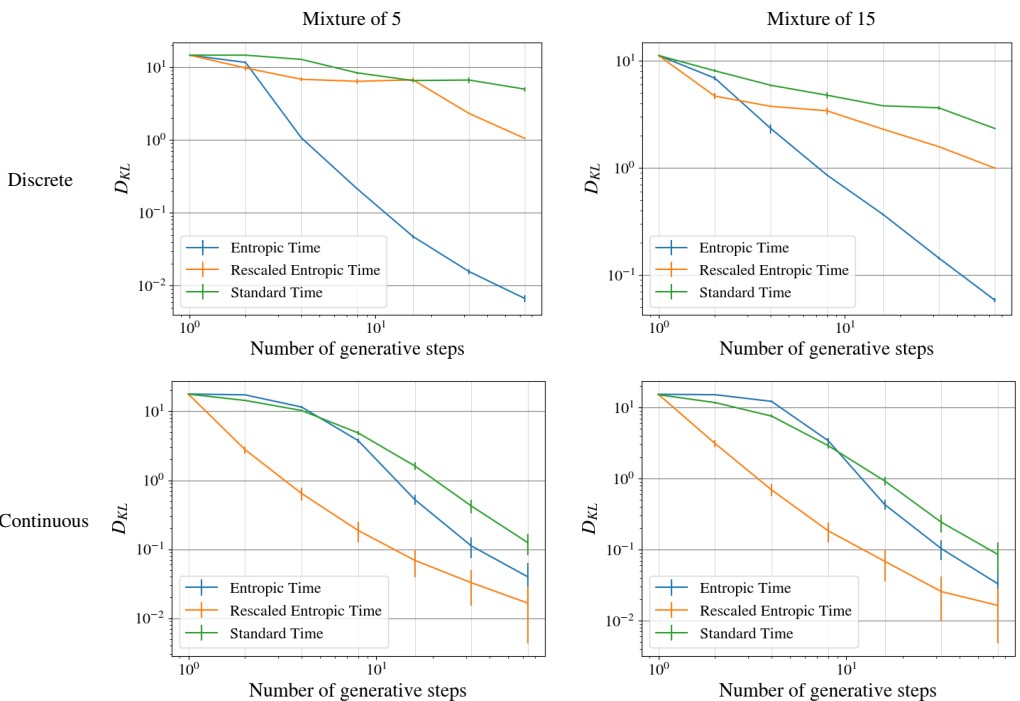

Figure 5: Kullback–Leibler divergence against the number of generative steps for different time parameterizations for a mixture of data points (discrete) and Gaussians (continuous).

mixture of Gaussians (continuous case). In both cases, the data had a mean of zero and a standard deviation of one. For sample generation, we used the stochastic DDIM (Song et al., 2022). Results are given in figure 5.

For the discrete case, the performance was measured by creating nonoverlapping bins around data points, $[a_i - \epsilon, a_i + \epsilon]$, and calculating the Kullback-Leibler divergence between the initial distribution and the binned distribution ($p_{bin}(a_i) =$ probability of a generated sample ending up in the $i$-th bin). A variance-preserving SDE and EDM SDE were used for our experiments. Datapoints were randomly initialized and Kullback-Leibler divergence was estimated $10^2$ times using $10^4$ different paths, so the mean and variance of the KL estimate could be obtained.

For the continuous case, the performance was measured by estimating the SDE-generated distribution using Gaussian kernel density estimation (with a standard deviation of $10^{-2}$) and then evaluating the KL divergence using Monte-Carlo methods with $10^3$ samples. Similarly to the discrete case, the KL divergence was estimated $10^2$ times using $10^4$ different paths to estimate the SDE-generated distribution.

### H.3 CIFAR10, FFHQ, ImageNet

For ImageNet-$64$, we used EDM2-S and EDM2-L models and for ImageNet-512, we used EDM2-XS and EDM2-XXL models provided by Karras et al. (2024). For CIFAR10 and FFHQ, we used unconditional VP models provided by Karras et al. (2022). For generating samples, we used the stochastic and deterministic DDIM (Song et al., 2022). To compare performance between different runs, we used the FID (Heusel et al., 2017) and, for ImageNet, FD-DINOv2 (Oquab et al., 2023; Stein et al., 2023) scores provided by the Karras et al. (2024) implementation. We used FD-DINOv2 as it correlates better with human preferences (Stein et al., 2023). We used implemantations provided by `https://github.com/NVlabs/edm`, for CIFAR10 and FFHQ, and `https://github.com/NVlabs/edm2`, for ImageNet. We generated $50000$ images and compared them against pre-computed reference statistics. All reported results are from a single (first) run. Class labels for ImageNet were drawn from a uniform distribution.

For all data sets, entropy and rescaled entropy were calculated using an estimation of squared error using equation 13. The squared error was estimated at 128 time points according to the EDM schedule ($\rho = 7$, $\sigma_{min} = 0.002$, $\sigma_{max} = 80$) using the Monte-Carlo method with 1024 samples at each timestep. In order to obtain (rescaled) entropy, any numerical integration technique should work. We decided on the simplest one, taking the difference in time steps, multiplying it by the derivative, and cumulatively summing it up to a time point $t$.

We decided on 128 time points by comparing it to 512 time points for CIFAR10 and FFHQ, and noticing no perceivable difference in the final entropy curves. Also, rescaled entropy was calculated for ImageNet-64 with both network sizes, $S$ and $L$, and there was no significant difference between them, as expected (since it depends only on a forward process and the initial data distribution). Therefore, we used the smallest models to estimate entropic quantities. Regarding a spectral rescaled entropy, 10000 images were used to estimate amplitudes of different frequencies.

As already stated in the main text, the entropic time generated blurry images and was not used for further comparison. An example of images generated with the deterministic DDIM solver using the entropic schedule over 64 steps, with the EDM2-L model, is given in figure 6. Results obtained for CIFAR10 and FFHQ are given in tables 3 and 1, respectively. Results for ImageNet-64 are given in table 4. Examples of generated images for ImageNet-64 using the EDM and rescaled entropy schedules are given in figures 8 and 9. For the sake of completeness, we include FID and FD-DINOv2 scores for models optimized for FID scores and models optimized for DINO scores in tables 5 and 6, respectively. We observe some interesting behavior of these results, such as DINO-optimized models giving better FID than FID-optimized ones for NFE=16. Also, we can see that the FID score can go up while the DINO score steadily decreases for DINO-optimized models. This shows that those two metrics asses and value vastly different properties of generated images. In addition, we notice that our DINO results are comparable to the results provided in Karras et al. (2024) obtained using Heun second-order solver. Figure 7 show how the number of function evaluations affect the generated images when using EDM and rescaled entropic schedules. Examples of generated images for ImageNet-512 using the EDM and rescaled entropy schedules with stochastic DDIM are given in figures 10, 11, and 12, while Examples of generated with deterministic DDIM are given in figures 13, 14, and 15.

Table 3: FID scores for different sampling schedules on CIFAR10 $32 \times 32$

| Solver | Schedule | FID ↓ | | |
|---|---|---|---|---|
| | | NFE=16 | NFE=32 | NFE=64 |
| Stochastic DDIM | EDM | 33.30 | 13.76 | 6.36 |
| | Rescaled Entropy | 20.07 | 8.44 | 4.65 |
| | Spectral Rescaled Entropy | **19.77** | **8.28** | **4.47** |
| Deterministic DDIM | EDM | 9.06 | 4.18 | 2.77 |
| | Rescaled Entropy | 6.07 | 3.30 | 2.52 |
| | Spectral Rescaled Entropy | **5.95** | **3.24** | **2.51** |

Table 4: FID and FD-DINOv2 scores for different sampling schedules for ImageNet-64

| Solver | Network | Schedule | FID ↓ | | | FD-DINOv2 ↓ | | |
|---|---|---|---|---|---|---|---|---|
| | | | NFE=16 | NFE=32 | NFE=64 | NFE=16 | NFE=32 | NFE=64 |
| Stochastic DDIM | EDM2-S | EDM | 20.03 | 8.18 | 3.81 | 263.60 | 135.67 | 86.32 |
| | | Rescaled Entropy | 11.69 | 4.95 | 2.75 | 194.02 | 109.55 | 81.25 |
| | | Spectral Rescaled Entropy | **11.46** | **4.76** | **2.70** | **193.81** | **109.16** | **79.75** |
| | EDM2-L | EDM | 22.60 | 9.46 | 4.44 | 284.74 | 141.70 | 79.86 |
| | | Rescaled Entropy | 13.56 | 5.59 | 3.06 | 208.27 | 108.31 | 72.06 |
| | | Spectral Rescaled Entropy | **13.46** | **5.51** | **2.99** | **207.76** | **106.42** | **70.37** |
| Deterministic DDIM | EDM2-S | EDM | 5.00 | 2.49 | 1.90 | 128.25 | 99.64 | **92.88** |
| | | Rescaled Entropy | **3.46** | 2.15 | **1.77** | **117.26** | **98.28** | 93.34 |
| | | Spectral Rescaled Entropy | 3.54 | **2.12** | 1.80 | 118.50 | 99.01 | 95.14 |
| | EDM2-L | EDM | 5.49 | 2.55 | 1.82 | 120.35 | 84.57 | **74.87** |
| | | Rescaled Entropy | **3.63** | **2.09** | 1.65 | **104.88** | **81.98** | 75.87 |
| | | Spectral Rescaled Entropy | 3.75 | **2.09** | **1.61** | 106.52 | 82.64 | 75.16 |

Table 5: FID and FD-DINOv2 scores for different sampling schedules for ImageNet-512 for models optimized for FID scores

| Solver | Network | Schedule | FID ↓ | | | FD-DINOv2 ↓ | | |
|---|---|---|---|---|---|---|---|---|
| | | | NFE=16 | NFE=32 | NFE=64 | NFE=16 | NFE=32 | NFE=64 |
| Stochastic DDIM | EDM2-XS | EDM | 32.31 | 10.01 | 4.98 | 419.27 | 199.69 | 131.94 |
| | | Rescaled Entropy | **13.64** | **4.98** | **3.80** | **280.82** | **154.24** | **124.07** |
| | EDM2-XXL | EDM | 30.39 | 8.80 | 3.81 | 337.23 | 127.69 | 68.97 |
| | | Rescaled Entropy | **13.38** | **3.83** | **2.60** | **186.68** | **82.47** | **57.41** |
| Deterministic DDIM | EDM2-XS | EDM | 10.42 | 4.81 | 3.83 | **212.78** | **154.41** | **137.68** |
| | | Rescaled Entropy | **7.57** | **4.44** | **3.75** | 222.15 | 161.79 | 142.22 |
| | EDM2-XXL | EDM | 9.68 | 3.47 | 2.41 | 137.35 | 81.26 | **65.75** |
| | | Rescaled Entropy | **5.91** | **2.78** | **2.14** | **125.35** | **79.80** | 66.75 |

Table 6: FID and FD-DINOv2 scores for different sampling schedules for ImageNet-512 for models optimized for DINO scores

| Solver | Network | Schedule | FID ↓ | | | FD-DINOv2 ↓ | | |
|---|---|---|---|---|---|---|---|---|
| | | | NFE=16 | NFE=32 | NFE=64 | NFE=16 | NFE=32 | NFE=64 |
| Stochastic DDIM | EDM2-XS | EDM | 17.14 | 8.76 | 7.34 | 294.25 | 149.91 | 107.00 |
| | | Rescaled Entropy | **7.76** | **6.39** | **6.43** | **182.11** | **109.68** | **97.10** |
| | EDM2-XXL | EDM | 15.94 | 7.36 | 6.06 | 218.10 | 95.21 | 60.79 |
| | | Rescaled Entropy | **6.84** | **5.02** | **5.03** | **108.16** | **57.05** | **46.75** |
| Deterministic DDIM | EDM2-XS | EDM | 7.16 | 5.97 | 5.80 | **156.46** | **115.94** | 107.05 |
| | | Rescaled Entropy | **6.00** | **5.31** | **5.40** | 157.32 | 116.52 | **106.84** |
| | EDM2-XXL | EDM | 5.48 | 4.10 | 4.03 | 79.56 | 52.60 | 46.27 |
| | | Rescaled Entropy | **3.81** | **3.54** | **3.70** | **68.36** | **48.26** | **43.83** |

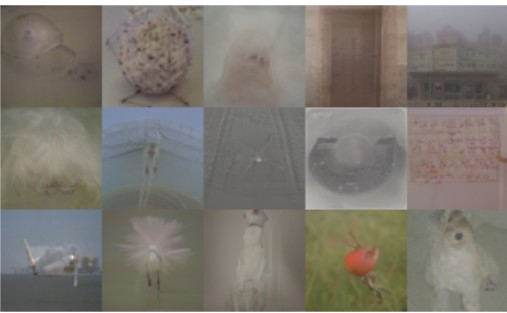

Figure 6: Images generated with the deterministic DDIM solver using the non-rescaled entropic schedule over 64 steps, with the EDM2-L model. It is clear from these images that rescaling is crucial in the continuous regime, probably due to the divergence of the differential entropy at $t \to 0$.

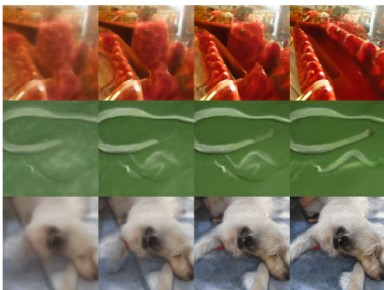 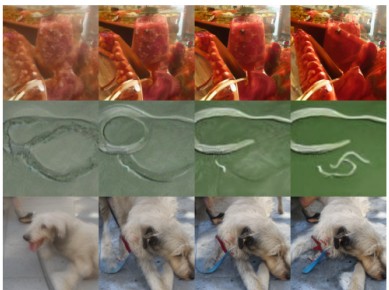

(a) EDM                  (b) Rescaled Entropy

Figure 7: Comparison of generated images using EDM and rescaled entropic schedules with the same random seed. Images were generated using deterministic DDIM with NFE = 8, 16, 32, and 64.

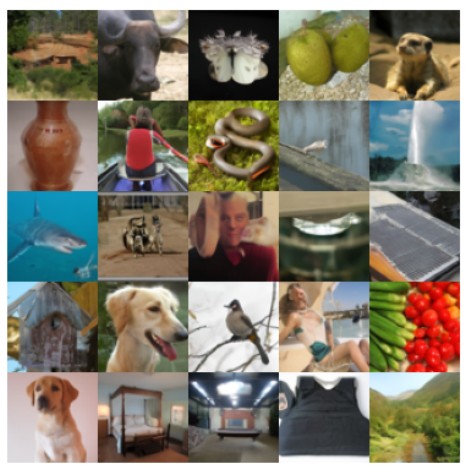

(a) EDM                  (b) Rescaled Entropy

Figure 8: Images generated with the stochastic DDIM solver using the EDM schedule (left) and rescaled entropic schedule (right) over 64 steps, with the EDM2-S model.

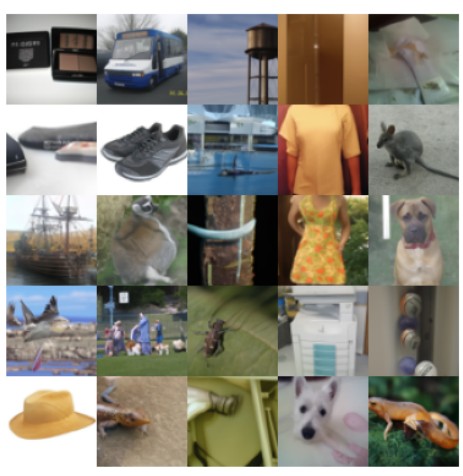 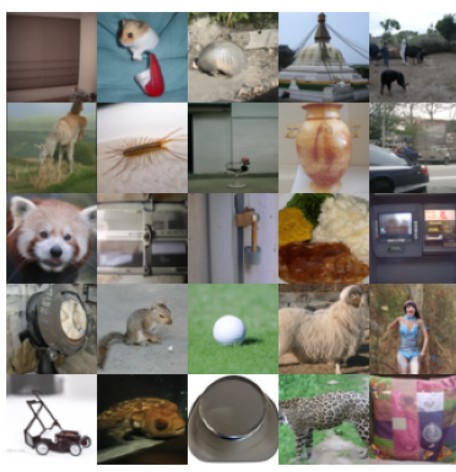

(a) EDM                  (b) Rescaled Entropy

Figure 9: Images generated with the stochastic DDIM solver using the EDM schedule (left) and rescaled entropic schedule (right) over 64 steps, with the EDM2-L model.

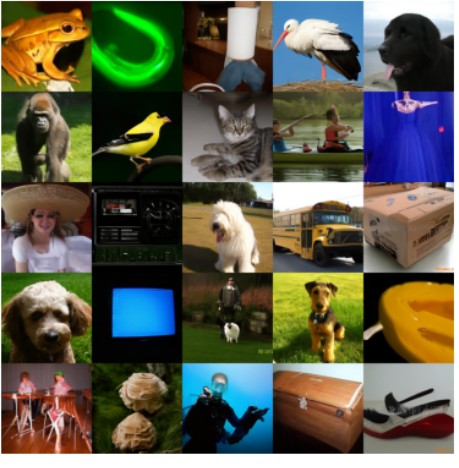 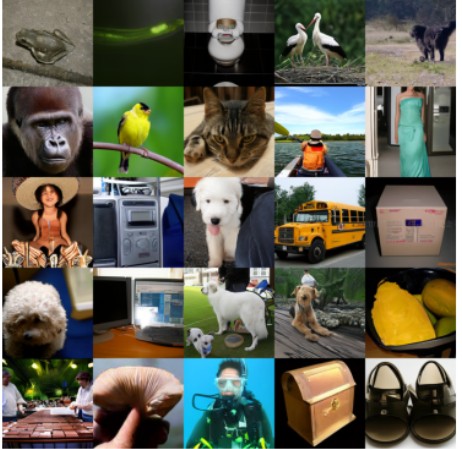

(a) EDM                                    (b) Rescaled Entropy

Figure 10: Images generated with the stochastic DDIM solver using the EDM schedule (left) and rescaled entropic schedule (right) over 64 steps, with the EDM2-XS DINO-optimized model.

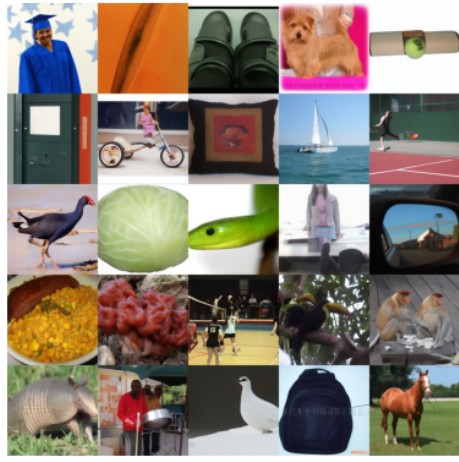 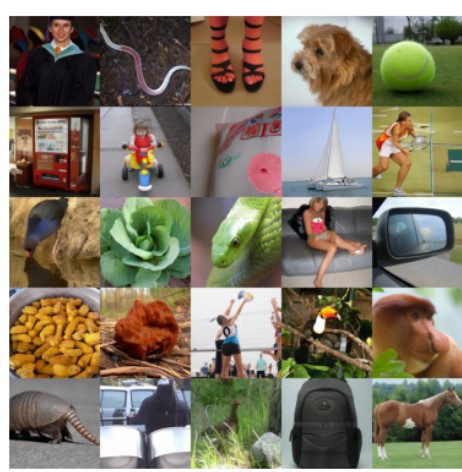

(a) EDM                                    (b) Rescaled Entropy

Figure 11: Images generated with the stochastic DDIM solver using the EDM schedule (left) and rescaled entropic schedule (right) over 64 steps, with the EDM2-XXL DINO-optimized model.

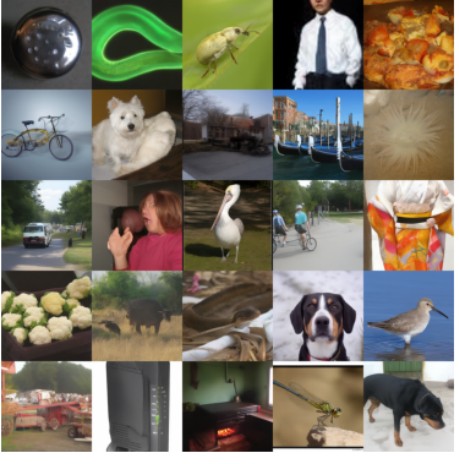 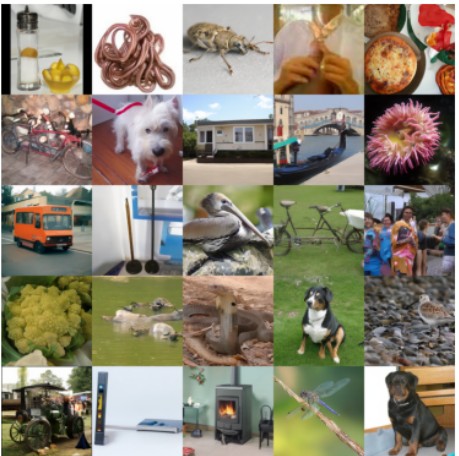

(a) EDM                                    (b) Rescaled Entropy

Figure 12: Images generated with the stochastic DDIM solver using the EDM schedule (left) and rescaled entropic schedule (right) over 64 steps, with the EDM2-XXL FID-optimized model.

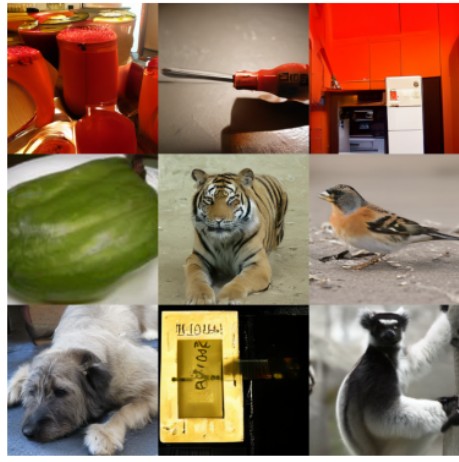 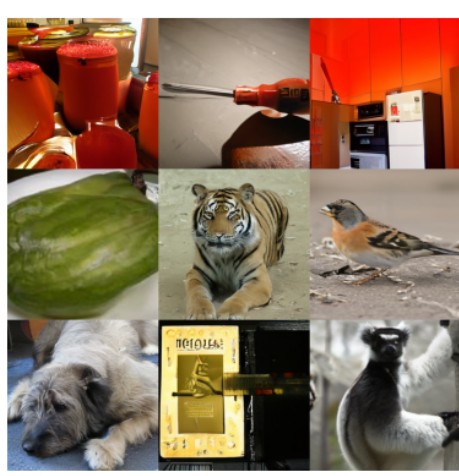

(a) EDM                                    (b) Rescaled Entropy

Figure 13: Images generated with the deterministic DDIM solver using the EDM schedule (left) and rescaled entropic schedule (right) over 64 steps, with the EDM2-XS DINO-optimized model.

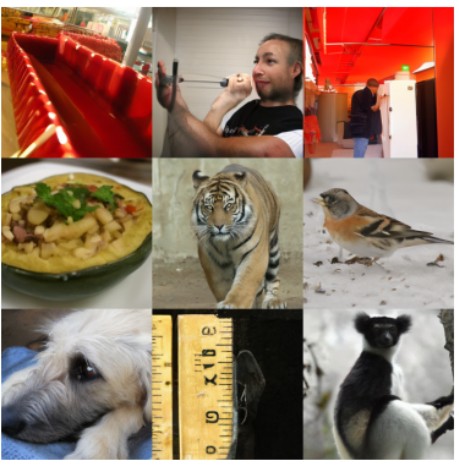 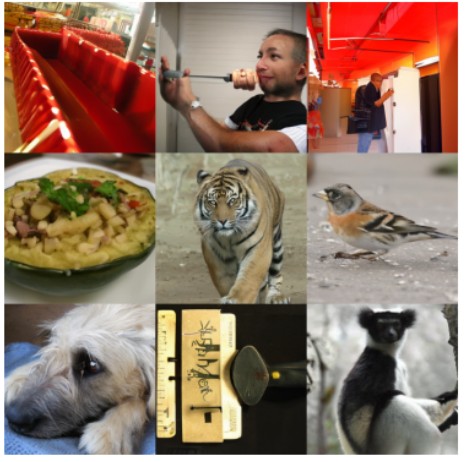

(a) EDM                                  (b) Rescaled Entropy

Figure 14: Images generated with the deterministic DDIM solver using the EDM schedule (left) and rescaled entropic schedule (right) over 64 steps, with the EDM2-XXL DINO-optimized model.

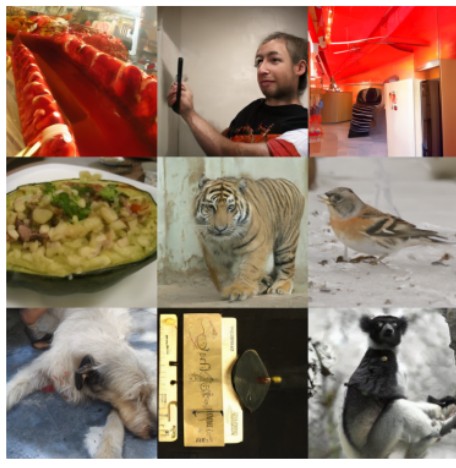 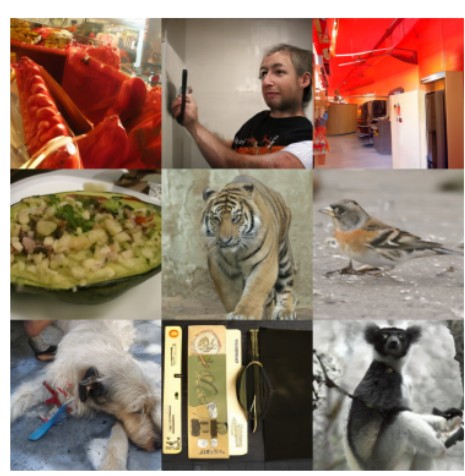

(a) EDM                                  (b) Rescaled Entropy

Figure 15: Images generated with the deterministic DDIM solver using the EDM schedule (left) and rescaled entropic schedule (right) over 64 steps, with the EDM2-XXL FID-optimized model.

