# OpenReview forum: "Entropic Time Schedulers for Generative Diffusion Models"
_NeurIPS.cc/2025/Conference — NeurIPS 2025 poster_

### Official Review · Reviewer_8dRB · 2025-06-30

**Clarity:** 4
**Significance:** 4
**Originality:** 3
**Rating:** 5
**Confidence:** 4

**Summary:**

This paper proposes an effective noise scheduling method at inference time (generation time) for the continuous-time denoising diffusion generative model family, aiming to achieve high sample quality, even with a low number of function evaluations (NFEs), for both discrete and continuous random variables.

Inspired by Sabour et al. (2024), the authors aim to find a new time parameterization under which linear-in-time discretization yields the best sample quality. To achieve this, the goal is then to design a change of time function $\phi(t)$, which must be invertible — hence, continuous and monotonically increasing.

The core idea is to use information transfer — or, more precisely, the mutual information between clean data $X_0$ and noisy data $X_t$ — to guide the time change. The authors propose a schedule where the rate of information transfer (i.e., its time derivative) increases linearly. Intuitively, this means that the time scheduling should be designed such that the information gain at each discretization point is maximized equally, leading to better sample quality.

Specifically, instead of directly using mutual information, the authors define the time change function as $\phi(t) = H[X_0|X_t]$, and refer to this as the entropic time schedule (Theorem 5.1). Note that the mutual information satisfies $I(X_0; X_t) = H[X_0] - H[X_0|X_t]$ decreases monotonically as $H[X_0]$ is fixed while the forward process is a linear diffusion. As a result, $H[X_0|X_t]$ increases monotonically over time. This ensures that $\phi$ is invertible and suitable for change-of-variable techniques.

Although computing $H[X_0|X_t]$ exactly is technically challenging, the authors show that only the derivative of $\phi$ is required for sampling (Definitions 4.1 and 4.2), and this derivative can be approximated. In particular, they express the rate of the entropic time schedule in terms of the denoising loss with respect to the true posterior (Equation 12) and propose a method to approximate it by plugging the learned score function into Equation 12 (Section 5.2).

The paper also points out that in the case of continuous random variables, the entropic time schedule may become unstable as $t \rightarrow 0$, since $H[X_0|X_t] \rightarrow -\infty$. To address this, the authors propose a rescaled time schedule (Theorem 5.1).

Additionally, they demonstrate that the proposed change-of-time function is invariant to the distribution of $X_0$ when the forward diffusion is fixed and can be applied across different bases.

Finally, they empirically validate their method on both discrete and continuous random variables, showing that it is effective in improving sample quality.

**Questions:**

While I understand the goal of finding a new time parameterization under which linear-in-time discretization yields the best sample quality, I am not fully convinced why linear time, in particular, is necessary. My question stems from a lack of familiarity with Sabour et al. (2024), which the paper builds upon. Suppose the key idea is to manage the rate of information gain. Would it not be more effective to maximize the information gain rate at each discretization step while minimizing it in regions that are not discretized? I’m not sure if I missed it, but is there any rationale — theoretical or empirical — for favoring a constant information gain rate over a more adaptive one?


In addition, the paper acknowledges that the plug-in approximation of Equation 12 introduces error. However, in lines 195–211, the analysis focuses only on how the ground-truth conditional entropy $H[X_0 | X_t]$ relates to the approximation error. Would it not be more appropriate to analyze the estimation bias introduced by using the learned score function in Equation 12, especially since this directly affects the accuracy of the entropic time schedule?


Moreover, do you believe that the proposed principle could also benefit training, not just sampling? It would be interesting to see whether adjusting time sampling during training (based on the current estimate) could further enhance model performance.


Lastly, I noticed that Kingma et al. (2021) also provide relevant discussion regarding time reparameterization, particularly in relation to Definitions 4.1 and 4.2. While perhaps not a core baseline, this work seems worth citing as a related reference.


Kingma et al. "Variational diffusion models." NeurIPS 2021

**Ethical Concerns:**

["NO or VERY MINOR ethics concerns only"]

**Final Justification:**

The revisions have further strengthened an already high-quality paper. While I believe a score of 5 is the most appropriate, I would have chosen 5.5 if such an option were available.

**Quality:**

3

**Strengths And Weaknesses:**

**Strengths**

I consider that the contribution of the paper is clear and important for several reasons.

The authors provide a compelling motivation for both the problem and the proposed solution. I believe the acceleration of the sample generation with diffusion-based generative models is still an important problem, and the paper proposes a solution based on information information-theoretic perspective. In particular, the authors articulate the core ideas in a way that is intuitive and accessible to readers, whether they are new to the topic or already familiar with the underlying concepts.

The theoretical derivations are concise and straightforward, and the overall presentation is clean and well-structured, which enhances readability and clarity.

**Weaknesses**

While the paper is technically sound and well-executed, it would be interesting to include further discussion on **why** **specifically linear-in-time discretization** under the proposed time parameterization leads to improved sample quality. This may (or may not) be relevant to the question stated below. This is not essential, but such insights could enrich the reader’s understanding.

---

> ### Author Rebuttal · Authors · 2025-07-30
>
> We value the detailed review and thoughtful feedback you provided. Our responses to your specific questions and the identified weaknesses are outlined in the following sections.
>
>
> ### Q1: While I understand the goal of finding a new time parameterization under which linear-in-time discretization yields the best sample quality, I am not fully convinced why linear time, in particular, is necessary. My question stems from a lack of familiarity with Sabour et al. (2024), which the paper builds upon. Suppose the key idea is to manage the rate of information gain. Would it not be more effective to maximize the information gain rate at each discretization step while minimizing it in regions that are not discretized? I’m not sure if I missed it, but is there any rationale — theoretical or empirical — for favoring a constant information gain rate over a more adaptive one?
>
> Our goal was to design a time parameterization in which each sampling step corresponds to a similar level of denoising difficulty. Since diffusion models progressively reduce uncertainty, we reasoned that a meaningful measure of step difficulty is the amount of remaining uncertainty about the data at a given point. This led us to use conditional entropy as a proxy: by constructing a time axis where each interval between solver steps spans the same amount of conditional entropy reduction, we aim to distribute the solver’s effort uniformly over the denoising process. This is the core motivation behind entropic time, where uniform discretization naturally aligns with consistent denoising effort per step.
>
> Regarding the linearity: in simple Gaussian toy models, the optimal sampling schedule for various noise schedules turns out to be linear in a reparameterized time variable, such as rescaled or entropic time. More generally, for any set of sampling time points, there always exists a reparameterization in which those points are equally spaced. The deeper question is whether a single function should generate all sampling schedules across different number of steps (e.g., entropic time vs. EDM time). While we do not know the answer, it aligns well with the core idea behind entropic time (constructing a time change under which entropy is linear) and is supported by analytical results in simple toy examples.
>
> ### Q2: The paper acknowledges that the plug-in approximation of Equation 12 introduces error. However, in lines 195–211, the analysis focuses only on how the ground-truth conditional entropy relates to the approximation error. Would it not be more appropriate to analyze the estimation bias introduced by using the learned score function in Equation 12, especially since this directly affects the accuracy of the entropic time schedule?
>
>
> We agree that analyzing the bias introduced by using the learned score function in Equation 12 is an important direction. Equation 16 partially addresses this by implying that the estimate using a learned score is always greater than or equal to the true conditional entropy production (there is a typo in the paper: the sentence should state that the right-hand side is our estimate of the conditional entropy production, not the left-hand side). This inequality provides a useful lower bound, but does not fully characterize the bias. While a deeper analysis of the estimation bias would be valuable, it is highly non-trivial in realistic settings where neither the exact score nor a reliable score-free entropy estimator is available. We consider this a promising topic for future theoretical investigation. If you believe this point deserves greater emphasis or clarification in the paper, we would be happy to revise the corresponding section accordingly.
>
> ### Q3: Do you believe that the proposed principle could also benefit training, not just sampling? It would be interesting to see whether adjusting time sampling during training (based on the current estimate) could further enhance model performance.
>
> We agree that using (rescaled) entropic time during training is a promising direction for future work. While this project focused on understanding the relationship between conditional entropy and diffusion models in the context of sampling efficiency, extending the principle to influence training dynamics is a natural next step. We plan to explore this idea in future work and will add a discussion of possible approaches in the discussion of the updated manuscript.
>
> ### Q4: Lastly, I noticed that Kingma et al. (2021) also provide relevant discussion regarding time reparameterization, particularly in relation to Definitions 4.1 and 4.2. While perhaps not a core baseline, this work seems worth citing as a related reference. (Kingma et al. "Variational diffusion models." NeurIPS 2021).
>
> Thank you for pointing out this relevant connection. Kingma et al. (2021) share conceptual similarities with our approach in exploring time reparameterization, but the goals and methods differ.
>
> Kingma et al. show that their continuous variational lower bound is invariant under time reparameterization, and they propose a learnable sampling schedule designed to minimize the variance of the training loss. In contrast, our focus is primarily on the sampling phase and on demonstrating the usefulness of entropy and related quantities for understanding and improving diffusion processes. Rather than learning a schedule during training, we construct the entropic time axis directly from the properties of the learned model. Moreover, our method only requires access to a trained score model, while theirs depends on a specific training loss.
>
> Interestingly, their continuous variational bound also connects to our framework: taking its expectation with respect to $x_0 \sim p_0(\cdot)$ yields the conditional entropy. This further supports the motivation for considering entropic time in the training phase of diffusion models and also shows the ubiquity of entropy and information-theoretic concepts in diffusion models. We will be sure to cite and discuss this connection more explicitly in the revised version.

---

### Official Review · Reviewer_HXTo · 2025-07-02

**Clarity:** 4
**Significance:** 3
**Originality:** 3
**Rating:** 5
**Confidence:** 5

**Summary:**

The paper tackles the task of finding better timestep schedules used during sampling of diffusion models. This is done by introducing *entropic time* which is a dataset-dependent time reparametrization of the diffusion process that ensures a constant rate of increase in the entropy of the denoising distribution $p(x_0 | x_t)$. The authors analyze this concept theoretically, demonstrating its independence of the initial SDE parameters, as well as its direct connection to the optimal schedule in the simple isotropic Gaussian setting. Experiments on both toy data as well as several standard image generation benchmarks demonstrates a clear benefit of using this parametrization over the commonly used schedules in practice.

**Questions:**

* Given the fact that this entropic time is only dataset-dependent, have the authors also explored the effects of using this parametrization as the noise schedule during the training of the diffusion model? I understand this requires a pretrained score model to compute accurately, but am curious whether a noisy estimation of it mid-training would suffice.
* How would one scale this idea to large scale diffusion models such as text-to-image and text-to-video models? Would it make sense to learn a class-conditioned (or text-conditioned) time parametrization?

**Ethical Concerns:**

["NO or VERY MINOR ethics concerns only"]

**Final Justification:**

The paper presents a theoretically motivated approach to optimize sampling schedules in order to speed up diffusion inference. The proposed approach is simple, elegant, and efficient to compute and the papers results are promising. As such, I recommend accepting this paper.

**Limitations:**

Yes.

**Paper Formatting Concerns:**

No major concerns.

**Quality:**

4

**Strengths And Weaknesses:**

**Strengths:**
* The paper is extremely well written and easy to follow.
* The core idea is simple, elegant, and well motivated. The paper also does a superb job explaining the connection between theoretical concepts and how to use them in practice.
* The proposed approach is training-free, and can easily be applied to any pretrained diffusion model.
* The experimental results are promising, demonstrating a significant boost in performance almost across the board.
* The limitation section is quite thorough, which is very appreciated.

**Weaknesses:**
* As discussed by the authors, the proposed entropic time parametrization seems to fail when used with higher-order solvers.
* As the case with all works that rely on optimizing the timestep schedule, this approach requires quite a few number of function evaluations to achieve best results and is not applicable to 1- or 2-step sampling scenarios.

**Minor nitpicks:**
* Figure 2 is very effective in showcasing the different schedules based on the radial frequency. I suggest also including a version of the same figure where the x-axis (time) is in logarithm scale to be able to better see the curves at the lower noise scales.
* Figures 7-11 are all using the stochastic DDIM sampler which causes different starting seeds to result in completely different images during sampling due to the different schedules. Including a few more figures that use the deterministic DDIM (similar to Figure 4) would be good to see.

---

> ### Author Rebuttal · Authors · 2025-07-30
>
> We appreciate your thorough review and constructive feedback. Please find below our responses addressing your specific questions and the weaknesses you highlighted.
>
> ### Q1: Given the fact that this entropic time is only dataset-dependent, have the authors also explored the effects of using this parametrization as the noise schedule during the training of the diffusion model? I understand this requires a pretrained score model to compute accurately, but am curious whether a noisy estimation of it mid-training would suffice.
>
> We agree that using (rescaled) entropic time as a training-time noise schedule is a promising direction. While our current project focused on exploring the connection between conditional entropy and diffusion models in the context of sampling efficiency, we plan to investigate the training-time use of entropic time in future work. As Equation 11 shows, entropy depends only on the squared norm of the score, not its direction. This suggests that even a semi-trained model might provide a sufficiently accurate estimate of the (rescaled) entropy.
>
> ### Q2: How would one scale this idea to large scale diffusion models such as text-to-image and text-to-video models? Would it make sense to learn a class-conditioned (or text-conditioned) time parametrization?
>
> We believe our approach is readily scalable to large models such as text-to-image or text-to-video diffusion models, especially since the entropic schedule is computed only once and reused at inference time. That said, using entropic time for training might have a larger effect and requires further investigation. We also find the idea of class-conditioned entropy compelling. While estimating class-conditioned entropy across broad categories may be feasible, conditioning on individual prompts likely would not be practical due to their rarity and the need for on-the-fly estimation (in the case of prompts). We plan to explore these directions in future work.
>
>
> ### W2: As the case with all works that rely on optimizing the timestep schedule, this approach requires quite a few number of function evaluations to achieve best results and is not applicable to 1- or 2-step sampling scenarios.
>
> Thank you for raising this point. Our goal was not to target consistency models or distilled models with extremely few sampling steps. It remains an open question whether an entropic time parametrization could benefit such approaches. We find this idea interesting and might include a discussion of it in the revised version.
>
> ### W3: Figure 2 is very effective in showcasing the different schedules based on the radial frequency. I suggest also including a version of the same figure where the x-axis (time) is in logarithm scale to be able to better see the curves at the lower noise scales.
>
> Thank you for the suggestion. We agree that applying a logarithmic scaling to the x-axis could offer better insight into the behavior of the schedules at lower noise levels. We will include this version of the figure in the revised paper.
>
> ### W4: Figures 7–11 are all using the stochastic DDIM sampler which causes different starting seeds to result in completely different images during sampling due to the different schedules. Including a few more figures that use the deterministic DDIM (similar to Figure 4) would be good to see.
>
> We agree that including more deterministic DDIM samples would help better isolate the effect of different schedules. We will include such figures in the updated version of the paper.

---

> > ### Comment · Reviewer_HXTo · 2025-08-06
> >
> > I appreciate the author's response and I look forward to future work incorporating similar ideas in diffusion training. My concerns have been addressed. I keep my rating and recommend to accept this paper.

---

### Official Review · Reviewer_3sWm · 2025-07-05

**Clarity:** 2
**Significance:** 2
**Originality:** 1
**Rating:** 2
**Confidence:** 4

**Summary:**

While the paper presents an interesting perspective by reparameterizing diffusion time based on conditional entropy, the proposed method lacks sufficient methodological depth. The core idea revolves around a theoretical reinterpretation of time scheduling rather than the development of a novel algorithmic mechanism. No new training framework, architectural design, or solver adaptation is introduced to complement the entropic schedule. Moreover, the scheduling function is computed post hoc from the training loss and is not jointly optimized with the model, limiting its potential. The method is also tightly coupled to first-order solvers and fails to generalize to higher-order methods, which further restricts its applicability. Finally, the spectral rescaled entropy schedule is introduced heuristically without theoretical grounding or ablation analysis, reducing the overall rigor and innovation of the approach.

**Questions:**

How do you justify the use of conditional entropy over mutual information as the time-rescaling metric, especially considering both are involved in Eq. (7)?

**Ethical Concerns:**

["NO or VERY MINOR ethics concerns only"]

**Limitations:**

The paper does not propose any new architectural modification or algorithmic mechanism beyond reparameterizing the time axis based on estimated conditional entropy.

**Quality:**

1

**Strengths And Weaknesses:**

The paper does not propose any new architectural modification or algorithmic mechanism beyond reparameterizing the time axis based on estimated conditional entropy. The practical implementation essentially relies on post hoc schedule computation without integration into the model training pipeline.

---

> ### Author Rebuttal · Authors · 2025-07-30
>
> We thank the reviewer for their feedback. However, we respectfully believe that several core contributions of the paper have been overlooked. Below, we address the major concerns.
> ### On Novelty and Theoretical Contribution
> Our work introduces a novel and theoretically grounded framework for rescheduling diffusion time using conditional entropy. This includes:
>
> - The formulation of entropic time and rescaled entropic time as new families of time reparameterizations, grounded in information-theoretic and stochastic process theory.
> - Theoretical results establishing that these schedules are invariant under time reparameterization of the forward process (Theorems 5.1 and 5.2).
> - A practical, tractable estimator of entropic time (Equation 13).
>
> ### On Practical Utility and Empirical Performance
> Our method is not only theoretically motivated, it is also practically useful. In all experiments, the proposed rescaled entropic time schedule significantly outperforms the standard EDM schedule. We observe improvements in FID and consistency across a range of datasets and solvers (Tables 1–4).
>
> ### On Post Hoc Schedule Optimization
> We believe that the simplicity of our approach, which does not require retraining and is agnostic to the architecture of the model, is a key strength of the method. While we agree that integrating our schedule into the training pipeline is a valuable direction for future work, our current focus is on improving sampling efficiency at inference time, which remains a critical challenge in modern generative modeling. For instance, the EDM schedule is highly optimized for inference performance. Identifying better sampling schedules can significantly enhance the practical utility of diffusion models.
>
> ### On Solver Generality
> While our method performs well with first-order solvers, we observed weaker results with second-order methods. As we explain, this is likely due to a mismatch in temporal structure: the current entropy-based schedule only considers uncertainty at a single time point, whereas second-order solvers depend on both current and future states. We believe this limitation can be addressed, and that extending the entropic time framework to account for this “look-ahead” behavior could enable similar benefits for higher-order solvers as well.
>
>
> ### On the Spectral Rescaled Entropy Schedule
> The spectral rescaled entropy schedule, introduced in Section 5.4, is meant to illustrate that our framework accommodates different orthonormal bases beyond the standard pixel-space representation. We chose the spectral basis as it is a natural example that aligns with practitioners' intuitions about frequency content and information distribution. This variant is motivated by the goal of equalizing the contribution of conditional entropy across frequency bands, thereby distributing denoising effort more uniformly.
>
> We agree that this component could benefit from additional clarification, and we are happy to expand on its motivation and include a more detailed explanation in the final version.
>
> ### On Conditional Entropy vs. Mutual Information
> Using mutual information would lead to the wrong monotonicity, as it strictly decreases in time, therefore violating what constitutes a valid change of time. One could propose to instead use $-I[\mathbf{x}_0;\mathbf{x}_t]$. However, this is entirely equivalent to $H[\mathbf{x}_0|\mathbf{x}_t]$ as a time change function since it only shifts $\phi(t)$ by an irrelevant constant factor (remark after definition 4.2.).
>
> ### Summary
> We believe the paper offers a novel and theoretically meaningful contribution to the study of diffusion models. It introduces:
>
>
> - A principled framework for designing entropy-based schedules.
> - Theoretical results demonstrating invariance and structure.
> - Empirical evidence showing improved sampling quality across datasets.
>
> We hope that this clarification will lead to a reconsideration of the originality and significance of our work.

---

### Official Review · Reviewer_hpGc · 2025-07-06

**Clarity:** 3
**Significance:** 3
**Originality:** 4
**Rating:** 5
**Confidence:** 5

**Summary:**

This paper proposes Entropic Time Schedulers, a diffusion sampling scheduler that uses $H[x_0|x_t]$ as the time parametrization. The authors started from the motivating observation from Sabour et al. 2024 that even a simple data distribution such as $\mathcal{N}(0, c^2I)$, the optimal  sampling schedule has a parameter $c$, which depends on the data distribution. More importantly, the authors draw inspiration from Sabour et al. 2024 to think of the time scheduler as a transformation of time. Then the authors move on to propose the time parametrization of $H[x_0|x_t]$, which is the conditional entropy of the optimal denoising distribution $p(x_0|x_t)$ from Sabour et al. 2024. They formally derive the time derivative of $H[x_0|x_t]$, i.e. the entropy rate, and connect it with the training loss. They then introduce the concept of "entropic time" and "scaled entropic time", the latter of which is a stretched version of the former when integrating out the time for the entropy rate. To account for the unrealistic isotropic assumption in Sabour et al.'s optimal denoising distribution, the authors further propose spectral rescaled entropic time. In their experiments, the proposed entropic time schedulers show better performance than baselines in FFHQ and ImageNet.

**Questions:**

Practically, how is the scheduler constructed? How is the integral of the entropy rate calculated? What is the computational complexity?

The authors analyze the gap between Score Matching and Denoising Score Matching and the influence on the entropy rate, but how would this affect the sampling results?

**Ethical Concerns:**

["NO or VERY MINOR ethics concerns only"]

**Final Justification:**

This paper propose a diffusion sampling scheduler with a clean and principled derivation, providing new insights in this critical module of diffusion models. I recommend accepting this paper.

**Limitations:**

Yes.

**Quality:**

3

**Strengths And Weaknesses:**

\+ I like this paper a lot. It is very well written and very easy to follow.

\+ As I summarized above, the thought process is very clean and principled.

\+ All derivations are formal, though I didn't check the details of all the derivations and proofs.

\+ The authors also discussed the gap between ground-truth entropy rate and the one calculated from Denoising Score Matching.

\- This is probably a limitation that the authors didn't explicitly state: constructing the proposed scheduler seems to need to traverse the whole dataset for multiple times.

---

> ### Author Rebuttal · Authors · 2025-07-30
>
> Thank you for the detailed review and thoughtful feedback. Below we address specific questions and the pointed out weaknesses.
>
> ### Q1: Practically, how is the scheduler constructed? How is the integral of the entropy rate calculated? What is the computational complexity?
>
> We provide the relevant details in Appendix H.2, and we will improve visibility by including a pseudocode in the main paper. Specifically, entropy and rescaled entropy are estimated using the squared error in Equation 13, evaluated at 128 time points according to the EDM schedule ($\rho = 7$, $\sigma_{\min} = 0.002$, $\sigma_{\max} = 80$) via Monte Carlo with 1024 samples per step. The entropy integral is computed using a simple numerical integration scheme: we multiply the estimated entropy rate by the time step difference and apply cumulative summation. This method is computationally efficient and easily implementable.
>
> ### Q2: The authors analyze the gap between Score Matching and Denoising Score Matching and the influence on the entropy rate, but how would this affect the sampling results?
>
> This analysis is intended to clarify the theoretical relationship between entropy and different loss functions. Although not directly used in our sampling procedure, it reveals that the estimated conditional entropy using denoising score matching is always greater than or equal to the true conditional entropy. We included this primarily for theoretical insight.
>
> ### W1: Constructing the proposed scheduler seems to require traversing the whole dataset multiple times - is this a limitation?
>
> This is an insightful concern. However, we empirically observe that it is not necessary to traverse the entire dataset to compute the (rescaled) entropy. In practice, using a subset of the data suffices to obtain accurate estimates. A possible explanation is that the score function already captures global information about the dataset, allowing us to sample $x_0, x_t \sim p(x_0, x_t)$ from a subset without significant loss of accuracy. This observation is consistent with the estimation method described in Appendix H.

---

> > ### Comment · Reviewer_hpGc · 2025-08-05
> >
> > Thank the authors for their response, which resolved my confusions. Please incorporate these clarifications into the revision.

---

### Comment · Area_Chair_KRjU · 2025-08-05

Dear Reviewers, given the authors' response, if you have not done so, please raise any remaining questions and/or concerns in a timely fashion so the authors have a chance to reply.

I remind you that Reviewers must participate in discussions with authors before submitting "Mandatory Acknowledgement”.

Thank you for your work.

---

### Note · Authors · 2025-08-14

We would like to thank the ACs and reviewers for the time and effort they dedicated to evaluating our submission. We greatly appreciate the constructive feedback, including suggestions for additional figures, missing references, and clarifying questions. These not only guided improvements we will incorporate into the final draft, but also helped us sharpen the presentation of the work. We appreciate the recognition that the paper is clear and well-written, as we put significant effort into presenting a coherent story. While we regret that not all reviewers were able to participate in the discussion, we understand the difficulty given the large number of submissions this year. Overall, we are thankful for the thoughtful feedback and for the opportunity to improve our work through this process. We look forward to further developing this line of work and hope that our results will be useful to the community.

---

### Decision · Program_Chairs · 2025-09-17

**Decision:**

Accept (poster)

**Comment:**

Score-based generative models sample random variables approximately distributed as a target distribution by discretizing a backward SDE associated with a  forward diffusion process used to noise data points. It is known both in practice and theory that the performance of these procedures depend on the choice of the noise schedule used in the forward noising process.

The authors propose to introduce a new time parameterization of the SDE to improve sampling quality and motivate in particular the use of the conditional entropy under the optimal denoising distribution as a good time parameterization. They propose (Theorem 5) two time parameterizations referred to as entropic time and rescaled entropic time parameterizations.   Although  the conditional entropy can be challenging to estimate, the authors propose a way of estimating the conditional entropy rate from the loss function which allows a practical use of the time parameterizations.

Reviewers found the paper well written and the proposed approach well motivated, in particular the practical consequences of the  theoretical framework convinced most reviewers along with the experiments which show promising performance on classical datasets.

The authors answer to most concerns during the rebuttal and this contribution is of great practical interest for a large ML audience in a very active area of research. The authors are encouraged to include the comments provided during the discussion to improve the paper (including in the main paper a pseudocode and the practical implementation of the scheduler, discussing time reparameterization at training time, discussing the bias introduced by using the learned score function).